

# On the topology of solutions to random continuous constraint satisfaction problems

Jaron Kent-Dobias

Istituto Nazionale di Fisica Nucleare, Sezione di Roma I, Italy
ICTP South American Institute for Fundamental Research, São Paulo, Brazil
Instituto de Física Teórica, Universidade Estadual Paulista
"Júlio de Mesquita Filho", São Paulo, Brazil

jaron@ictp-saifr.org

## Abstract

We consider the set of solutions to $M$ random polynomial equations whose $N$ variables are restricted to the $(N-1)$-sphere. Each equation has independent Gaussian coefficients and a target value $V_0$. When solutions exist, they form a manifold. We compute the average Euler characteristic of this manifold in the limit of large $N$, and find different behavior depending on the target value $V_0$, the ratio $\alpha = M/N$, and the variances of the coefficients. We divide this behavior into five phases with different implications for the topology of the solution manifold. When $M = 1$ there is a correspondence between this problem and level sets of the energy in the spherical spin glasses. We conjecture that the transition energy dividing two of the topological phases corresponds to the energy asymptotically reached by gradient descent from a random initial condition, possibly resolving an open problem in out-of-equilibrium dynamics. However, the quality of the available data leaves the question open for now.



# 1  Introduction

Constraint satisfaction problems seek configurations that simultaneously satisfy a set of equations, and form a basis for thinking about problems as diverse as neural networks [1], granular materials [2], ecosystems [3], and confluent tissues [4]. All but the last of these examples deal with sets of inequalities, while the last considers a set of equality constraints. Inequality constraints are familiar in situations like zero-cost solutions in neural networks with ReLu activations and stable equilibrium in the forces between physical objects. Equality constraints naturally appear in the zero-gradient solutions to overparameterized smooth neural networks and in vertex models of tissues.

In problems ranging from toy models [5,6] to real deep neural networks [7–11], there is great interest in characterizing structure in the set of solutions, which can influence the behavior of algorithms trying to find them [12]. Here, we show how topological information about the set of solutions can be calculated in a simple problem of satisfying random nonlinear equalities. This allows us to reason about the connectivity and structure of the solution set. The topological properties revealed by this calculation yield surprising results for the well-studied spherical spin glasses, where a topological transition thought to occur at a threshold energy $E_{\text{th}}$ where marginal minima are dominant is shown to occur at a different energy $E_{\text{sh}}$. We conjecture that this difference resolves an outstanding problem with the out-of-equilibrium dynamics in these systems.

We consider the problem of finding configurations $\mathbf{x} \in \mathbb{R}^N$ lying on the $(N-1)$-sphere $\|\mathbf{x}\|^2 = N$ that simultaneously satisfy $M$ nonlinear constraints $V_k(\mathbf{x}) = V_0$ for $1 \le k \le M$ and some constant $V_0 \in \mathbb{R}$. The nonlinear constraints are taken to be centered Gaussian random functions with covariance

$$\overline{V_i(\mathbf{x})V_j(\mathbf{x}')} = \delta_{ij} f\left(\frac{\mathbf{x} \cdot \mathbf{x}'}{N}\right), \tag{1}$$

for some choice of function $f$. When the covariance function $f$ is polynomial, the $V_k$ are also polynomial, with a term of degree $p$ in $f$ corresponding to all possible terms of degree $p$ in the $V_k$. One can explicitly construct functions that satisfy (1) by taking

$$V_k(\mathbf{x}) = \sum_{p=0}^{\infty} \frac{1}{p!} \sqrt{\frac{f^{(p)}(0)}{N^p}} \sum_{i_1 \cdots i_p}^{N} J_{i_1 \cdots i_p}^{(k,p)} x_{i_1} \cdots x_{i_p}, \tag{2}$$

where the elements of the tensors $J^{(k,p)}$ are independently distributed unit normal random variables. The series coefficients of $f$ therefore control the variances of the random coefficients in the polynomials $V_k$. When $M = 1$, this problem corresponds to finding the level set of a spherical spin glass at energy density $E = V_0/\sqrt{N}$.

This problem or small variations thereof have attracted attention recently for their resemblance to encryption, least-squares optimization, and vertex models of confluent tissues [4, 13–24]. In each of these cases, the authors studied properties of the cost function

$$\mathscr{C}(\mathbf{x}) = \frac{1}{2} \sum_{k=1}^{M} \big[ V_k(\mathbf{x}) - V_0 \big]^2, \tag{3}$$

which achieves zero only for configurations that satisfy all the constraints. Introduced in Ref. [13], the existence of solutions and the geometric structure of the cost function were studied for the problem with linear $V_k$ in a series of papers [13–15] and later reviewed [17]. Some work on the equilibrium measure of the cost function with nonlinear $V_k$ was made in Ref. [16], and the problem was solved in Ref. [4]. Subsequent work has studied varied dynamics applied to the cost function, including gradient descent, Hessian descent, Langevin, stochastic gradient descent, and approximate message passing [18, 19, 21, 22]. Finally, some progress has been made on aspects of the geometric structure of the cost function with nonlinear $V_k$ [23, 24].

From the perspective of the cost function, the set of solutions looks like a network of flat canyons at the bottom of the cost landscape. Here we dispense with the cost function and study the set of solutions directly. This set can be written as

$$\Omega = \big\{ \mathbf{x} \in \mathbb{R}^N \mid \|\mathbf{x}\|^2 = N, V_k(\mathbf{x}) = V_0 \ \forall \ k = 1, \dots, M \big\}. \tag{4}$$

Because the constraints are all smooth functions, $\Omega$ is almost always a manifold without singular points.[1] We study the topology of the manifold $\Omega$ by computing its average Euler characteristic, a topological invariant whose value puts constraints on the manifold's structure. The topological phases determined by this measurement are distinguished by the size and sign of the Euler characteristic, and the distribution in space of its constituents.

In Section 2 we describe how to calculate the average Euler characteristic, how to interpret the results of that calculation, and what topological phases are implied. In Section 3 we examine some implications of these results for dynamic thresholds in the spherical spin glasses. Finally, in Section 4 we make some concluding remarks. Many of the details of the calculations in the middle sections are found in Appendices A–D.

## 2 The average Euler characteristic

### 2.1 Definition and derivation

The Euler characteristic $\chi$ of a manifold is a topological invariant [25]. It is perhaps most familiar in the context of connected compact orientable surfaces, where it characterizes the number of handles in the surface: $\chi = 2(1 - \#)$ for # handles. In higher dimensions it is more difficult to interpret, but there are a few basic intuitions. The Euler characteristic of the hypersphere is 2 in even dimensions and 0 in odd dimensions. In fact, the Euler characteristic

---

[1] The conditions for a singular point are that $0 = \frac{\partial}{\partial \mathbf{x}} V_k(\mathbf{x})$ for all $k$. This is equivalent to asking that the constraints $V_k$ all have a stationary point at the same place. When the $V_k$ are independent and random, this is vanishingly unlikely, requiring $NM + 1$ independent equations to be simultaneously satisfied. This means that different connected components of the set of solutions do not intersect, nor are there self-intersections, without extraordinary fine-tuning.

of an odd-dimensional manifold is always zero. The Euler characteristic of the union of two disjoint manifolds is the sum of the Euler characteristics of the individual manifolds, and that of the product of two manifolds is the product of the Euler characteristics. This means that a manifold made of many disconnected sphere-like components will have a large positive Euler characteristic. A manifold with many hyper-handles will have a large negative Euler characteristic. And no matter the Euler characteristic of a manifold, the Euler characteristic of its product with the circle $S^1$ is zero.

The canonical method for computing the Euler characteristic is to construct a complex on the manifold in question, which is a higher-dimensional generalization of a polygonal tiling. Then $\chi$ is given by an alternating sum over the number of cells of increasing dimension, which for 2-manifolds corresponds to the number of vertices, minus the number of edges, plus the number of faces. Morse theory offers another way to compute the Euler characteristic of a manifold $\Omega$ using the statistics of stationary points in a function $H : \Omega \to \mathbb{R}$ [26]. For functions $H$ without any symmetries with respect to the manifold, the surfaces of gradient flow between adjacent stationary points form a complex. The alternating sum over cells becomes an alternating sum over the count of stationary points of $H$ with increasing index, or

$$\chi(\Omega) = \sum_{i=0}^{N} (-1)^i \mathcal{N}_H(\text{index} = i). \tag{5}$$

Conveniently, we can express this sum as an integral over the manifold using a small variation on the Kac–Rice formula for counting stationary points [27,28]. Since the sign of the determinant of the Hessian matrix of $H$ at a stationary point is equal to its index, if we count stationary points including the sign of the determinant, we arrive at the Euler characteristic, or

$$\chi(\Omega) = \int_\Omega d\mathbf{x}\, \delta\big(\nabla H(\mathbf{x})\big) \det \text{Hess}\, H(\mathbf{x}). \tag{6}$$

When the Kac–Rice formula is used to calculate the total number stationary points, one must take pains to eliminate the sign of the determinant [29]. Here it is correct to preserve it.

We need to choose a function $H$ for our calculation. Because $\chi$ is a topological invariant, any choice will work so long as it does not have degenerate stationary points on the manifold, i.e., that it is a Morse function, and that it does not share some symmetry with the underlying manifold, i.e., that it satisfies the Smale condition. Because our manifold is random and has no symmetries, we can take a simple height function $H(\mathbf{x}) = \mathbf{x}_0 \cdot \mathbf{x}$ for some $\mathbf{x}_0 \in \mathbb{R}^N$ with $\|\mathbf{x}_0\|^2 = N$. We call $H$ a height function because when $\mathbf{x}_0$ is interpreted as the polar axis of a spherical coordinate system, $H$ gives the height on the sphere relative to the equator.

We treat the integral over the implicitly defined manifold $\Omega$ using the method of Lagrange multipliers. We introduce one multiplier $\omega_0$ to enforce the spherical constraint and $M$ multipliers $\omega_k$ for $k = 1, \ldots, M$ to enforce the $M$ constraints, resulting in the Lagrangian

$$L(\mathbf{x}, \boldsymbol{\omega}) = H(\mathbf{x}) + \frac{1}{2} \omega_0 \big( \|\mathbf{x}\|^2 - N \big) + \sum_{k=1}^{M} \omega_k \big( V_k(\mathbf{x}) - V_0 \big). \tag{7}$$

The integral over the solution manifold $\Omega$ in (6) becomes

$$\chi(\Omega) = \int_{\mathbb{R}^N} d\mathbf{x} \int_{\mathbb{R}^{M+1}} d\boldsymbol{\omega}\, \delta\big(\partial L(\mathbf{x}, \boldsymbol{\omega})\big) \det \partial\partial L(\mathbf{x}, \boldsymbol{\omega}), \tag{8}$$

where $\partial = [\frac{\partial}{\partial \mathbf{x}}, \frac{\partial}{\partial \boldsymbol{\omega}}]$ is the vector of partial derivatives with respect to all $N + M + 1$ variables. This expression is now in a form where standard techniques from the mean-field theory of disordered systems can be applied to average over the random constraint functions and evaluate the integrals to leading order in large $N$.

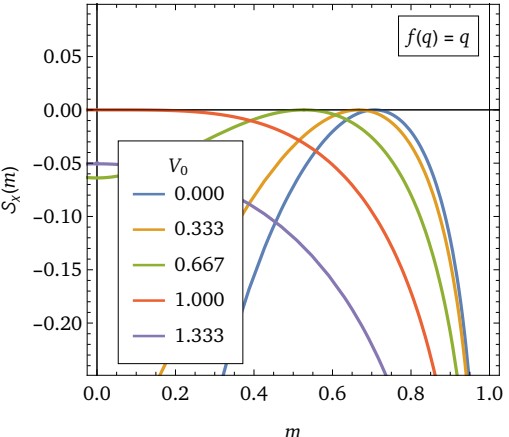 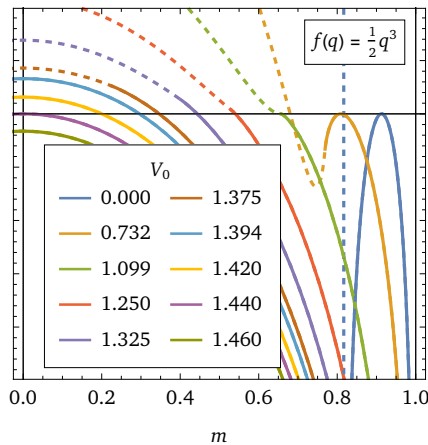

Figure 1: **Effective action for the Euler characteristic.** The action (10) as a function of $m = \frac{1}{N}\mathbf{x} \cdot \mathbf{x}_0$ for pure polynomial constraints and a selection of target values $V_0$. Dashed lines depict $\text{Re}\,\mathcal{S}_\chi$ when its imaginary part is nonzero. In both plots $\alpha = \frac{1}{2}$. **Left:** With linear functions there are two regimes. For small $V_0$, there are maxima at $m = \pm m_*$ where the action is zero, while for $V_0 > V_{\text{SAT}*} = 1$ the action is negative everywhere. **Right:** With nonlinear functions there are other possible regimes. For small $V_0$, there are maxima at $m = \pm m_*$ but the real part of the action is maximized at $m = 0$ where the action is complex. For larger $V_0 \geq V_{\text{on}} \simeq 1.099$ the maxima at $m = \pm m_*$ disappear. For $V_0 \geq V_{\text{sh}} \simeq 1.394$ larger still, the action becomes real everywhere. Finally, for $V_0 > V_{\text{SAT}} \simeq 1.440$ the action is negative everywhere.

Details of this calculation can be found in Appendix A. The result is the reduction of the average Euler characteristic to an integral over a single order parameter $m = \frac{1}{N}\mathbf{x} \cdot \mathbf{x}_0$ of the form

$$\overline{\chi(\Omega)} = \left(\frac{N}{2\pi}\right)^{\frac{1}{2}} \int dm\, g(m)\, e^{N\mathcal{S}_\chi(m)}, \tag{9}$$

where $g(m)$ is a prefactor of order $N^0$ and $\mathcal{S}_\chi(m)$ is an effective action defined by

$$\mathcal{S}_\chi(m) = -\frac{\alpha}{2}\left[\log\left(1 - \frac{f(1)}{f'(1)}\frac{1 + \frac{m}{R_m}}{1 - m^2}\right) + \frac{V_0^2}{f(1)}\left(1 - \frac{f'(1)}{f(1)}\frac{1 - m^2}{1 + \frac{m}{R_m}}\right)^{-1}\right] + \frac{1}{2}\log\left(-\frac{m}{R_m}\right). \tag{10}$$

Here we have introduced the ratio $\alpha = M/N$ between the number of equations and the number of variables, and $R_m$ is a function of $m$ given by

$$R_m \equiv \frac{-m(1 - m^2)}{2[f(1) - (1 - m^2)f'(1)]^2}\left[\alpha V_0^2 f'(1) + (2 - \alpha)f(1)\left(\frac{f(1)}{1 - m^2} - f'(1)\right)\right.$$
$$\left. + \alpha\sqrt{\frac{4V_0^2}{\alpha}f(1)f'(1)\left[\frac{f(1)}{1 - m^2} - f'(1)\right] + \left[\frac{f(1)^2}{1 - m^2} - (V_0^2 + f(1))f'(1)\right]^2}\right]. \tag{11}$$

The effective action (10) is plotted in Fig. 1 for a selection of parameters. To finish evaluating the integral by the saddle-point approximation, the action should be maximized with respect to $m$. If $m_*$ is such a maximum, then the resulting average Euler characteristic is $\overline{\chi(\Omega)} \propto e^{N\mathcal{S}_\chi(m_*)}$. In the next subsection we examine the maxima of $\mathcal{S}_\chi$ and their properties as the parameters are varied.

## 2.2 Features of the effective action

The order parameter $m$ is the overlap of the configuration $\mathbf{x}$ with the height axis $\mathbf{x}_0$. Therefore, the value $m$ that maximizes this action can be understood as the latitude on the sphere at which most of the contribution to the Euler characteristic is made.[2] The action $\mathcal{S}_\chi$ is extremized with respect to $m$ at $m = 0$ or at $m = \pm m_*$ for

$$m_* = \sqrt{1 - \frac{\alpha}{f'(1)}\big(V_0^2 + f(1)\big)}. \tag{12}$$

At these latter extrema, $\mathcal{S}_\chi(\pm m_*) = 0$. Zero action implies that $\overline{\chi(\Omega)}$ does not vary exponentially with $N$, and in fact we show in Appendix B that the contribution from these extrema is $1 + o(N^0)$ at $-m_*$ and $(-1)^{N-M-1} + o(N^0)$ at $+m_*$, so that their sum is 2 in even dimensions and 0 in odd dimensions. When these extrema exist and maximize the action, this result is consistent with the topology of an $N - M - 1$ sphere.

If this solution were always well-defined, it would vanish when the argument of the square root vanishes for

$$V_0^2 > V_{\text{SAT}*}^2 \equiv \frac{f'(1)}{\alpha} - f(1). \tag{13}$$

This corresponds precisely to the satisfiability transition found in previous work by a replica symmetric analysis of the cost function (3) [13–17]. However, the action is not clearly defined in the entire range $m^2 < 1$: it becomes complex in the region $m^2 < m_{\min}^2$ where

$$m_{\min}^2 \equiv 1 - \frac{f(1)^2}{f'(1)} \times \frac{V_0^2(1 + \sqrt{1-\alpha})^2 - \alpha f(1)}{4V_0^2 f(1) - \alpha[V_0^2 + f(1)]^2}. \tag{14}$$

When $m_*^2 < m_{\min}^2$, the solutions at $m = \pm m_*$ are no longer maxima of the action. This happens when the target value $V_0$ is larger than an onset value $V_{\text{on}}$ defined by

$$V_{\text{on}}^2 \equiv \frac{f(1)}{\alpha}\left(1 - \alpha + \sqrt{1-\alpha}\right). \tag{15}$$

Comparing this with the satisfiability transition associated with $m_*$ going to zero, one sees

$$V_{\text{on}}^2 - V_{\text{SAT}*}^2 = \frac{1}{\alpha}\left(f'(1) - f(1) - f(1)\sqrt{1-\alpha}\right). \tag{16}$$

If $f(q)$ is purely linear, then $f'(1) = f(1)$ and $V_{\text{on}}^2 > V_{\text{SAT}*}^2$, so the naïve satisfiability transition happens first. On the other hand, when $f(q)$ contains powers of $q$ strictly greater than 1, then $f'(1) \geq 2f(1)$ and $V_{\text{on}}^2 \leq V_{\text{SAT}*}^2$, so the onset happens first. In situations with mixed constant, linear, and nonlinear terms in $f$, the order of the transitions depends on the precise form of $f$.

---

[2]The order parameter $m$ may resemble the magnetization that appears in problems that have a signal or spike, where it gives the overlap of a configuration with the hidden signal. Here $\mathbf{x}_0$ is no signal, but a direction chosen uniformly at random and with no significance to the set of solutions. Here, if a feature of the action is present at some value $m$, it should be interpreted as indicating that, with overwhelming probability, typical configurations contributing to that feature have an overlap $m$ with a typical point in configuration space. For instance, for $m$ sufficiently close to 1, $\mathcal{S}_\chi(m)$ is always negative, which is a result of the absence of any stationary points contributing to the Euler characteristic at those overlaps. Given a random height axis $\mathbf{x}_0$, the nearest point to $\mathbf{x}_0$ on the solution manifold will be the absolute maximum of the height function, and therefore will contribute to the Euler characteristic. Hence the region of negative action in the vicinity of $m = 1$ implies there is a typical minimum distance between the solution manifold and a randomly drawn point in configuration space, and that it is vanishingly unlikely to draw a point in configuration space uniformly at random and find it any closer to the solution manifold than this. Other properties of the set of solutions could be studied by drawing $\mathbf{x}_0$ from an alternative distribution, like the Boltzmann distribution of the cost function, from the set of its stationary points, or from the solution manifold itself. While the value of the Euler characteristic would not change, the dependence of the effective action on $m$ would change.

Now we return to the extremum at $m = 0$. As for those at $\pm m_*$, the action evaluated at this solution is sometimes complex-valued and sometimes real-valued. For $V_0$ less than a shattering value $V_{\text{sh}}$ defined by

$$V_{\text{sh}}^2 \equiv \frac{f(1)}{\alpha}\left(1 - \frac{f(1)}{f'(1)}\right)\left(1 + \sqrt{1-\alpha}\right)^2, \tag{17}$$

the maximum at $m = 0$ is complex while for $V_0$ greater than this value the action is real. For purely linear $f(q)$, $V_{\text{sh}} = 0$ and the action at $m = 0$ is always real, though for $V_0^2 < V_{\text{SAT}*}^2$ it is a minimum rather than a maximum. Finally, there is another satisfiability transition at $V_0 = V_{\text{SAT}}$ corresponding to the vanishing of the effective action at the $m = 0$ solution, with $\mathcal{S}(0) = 0$. For a generic covariance function $f$ it is not possible to write an explicit formula for $V_{\text{SAT}}$, and we calculate it through a numeric root-finding algorithm.[3]

When $V_0^2 < V_{\text{sh}}^2$, the solution at $m = 0$ is difficult to interpret, since the action takes a complex value. Such a result could arise from the breakdown of the large-deviation principle behind the calculation of the effective action, or it could be the result of a negative Euler characteristic. To address this ambiguity, we compute also the average of the square of the Euler characteristic, $\overline{\chi(\Omega)^2}$, with details in Appendix C. This has the benefit of always being positive, so that the saddle-point approach to the calculation at large $N$ does not produce complex values even when $\overline{\chi(\Omega)}$ is negative. Under the restriction that $f(0) = 0$,[4] we identify three saddle points that could contribute to the value of $\overline{\chi(\Omega)^2}$: two at $\pm m_*$ where $\frac{1}{N}\log\overline{\chi(\Omega)^2} = \frac{1}{N}\log\overline{\chi(\Omega)} \simeq 0$, and one at $m = 0$ where

$$\frac{1}{N}\log\overline{\chi(\Omega)^2} = 2\,\mathrm{Re}\,\mathcal{S}_\chi(0), \tag{18}$$

which is consistent with $\overline{\chi(\Omega)^2} \simeq [\overline{\chi(\Omega)}]^2$. We therefore conclude that when the effective action is complex-valued, the average Euler characteristic is negative and its magnitude is given by the real part of the action.

Such a correspondence, which indicates that the 'annealed' calculation presented here is also representative of typical realizations of the constraints, is not always true. Sometimes the average squared Euler characteristic has alternative saddle points for which $\overline{\chi(\Omega)^2} \neq [\overline{\chi(\Omega)}]^2$, which implies that average properties will not be typical of most realizations. With our calculation of the average squared Euler characteristic, we can identify instabilities of the solution described above toward such replica symmetry breaking (RSB) solutions. The analysis of these instabilities can be found in Appendix C.2. We do not explore these RSB solutions here, except in the context of $M = 1$ and the spherical spin glasses in Section 3. However, in the phase diagrams of Figures 3 and 4 we shade the region where our calculation indicates that an instability is present.

## 2.3 Topological phases and their interpretation

The results of the previous section allow us to unambiguously define distinct topological phases, which differ depending on the presence or absence of the local maxima at $m = \pm m_*$,

---

[3]As a check of this calculation, the satisfiability threshold calculated here can be compared with that calculated using the zero-temperature limit of an equilibrium treatment of the cost function (3) made in Ref. [4] for the case where $f(q) = \frac{1}{2}q^2$ and $\alpha = \frac{1}{4}$. The authors estimate $V_{\text{SAT}} \simeq 1.871$, whereas this manuscript predicts $V_{\text{SAT}} = 1.867229\ldots$, a seeming inconsistency. However, the author of Ref. [4] indicated in private correspondence that this difference is explained by inaccuracy in the numeric PDE treatment of the FRSB equilibrium problem. Therefore, this manuscript is consistent with the previous work, but the agreement is not precise.

[4]This restriction is equivalent to having no random constant term in the constraint equations. It provides a simplification here because when it is present the replica symmetric (RS) description of this problem can have $q_0 > 0$, and $\overline{\chi(\Omega)^2} \neq [\overline{\chi(\Omega)}]^2$ always.

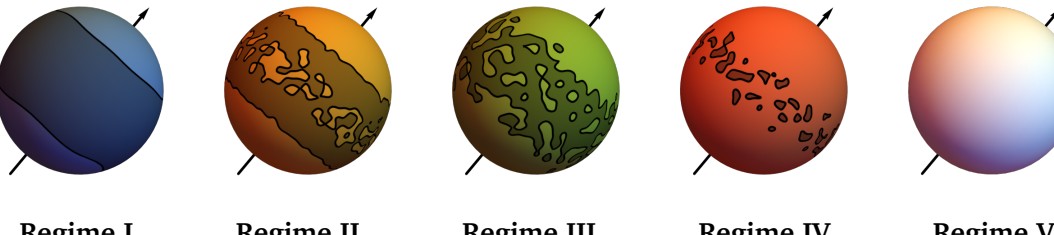

| Regime I | Regime II | Regime III | Regime IV | Regime V |

Figure 2: **Cartoons of the solution manifold in five topological regimes.** The solution manifold is shown as a shaded region, and the height axis $\mathbf{x}_0$ is a black arrow. In Regime I, the average Euler characteristic is consistent with a manifold with a single simply-connected component. In Regime II, holes occupy the equator but the temperate regions are topologically simple. In Regime III, holes dominate and the edge of the manifold is not necessarily simple. In Regime IV, disconnected components dominate. In Regime V, the manifold is empty.

on the presence or absence of the local maximum at $m = 0$, on the real or complex nature of this maximum, and finally on whether the action is positive or negative. Below we enumerate these regimes, which are schematically represented in Fig. 2.[5] It is not possible to definitively ascertain what structural features of the solution manifold lead to these average invariants, but we suggest a simplest interpretation consistent with the calculated properties.

**Regime I: $\overline{\chi(\Omega)} = 2$.** This regime is found when the magnitude of the target value $V_0$ is less than the onset $V_{\mathrm{on}}$ and $\mathrm{Re}\,\mathcal{S}(0) < 0$, so that the maxima at $m = \pm m_*$ exist and are the dominant contributions to the average Euler characteristic. Here, $\overline{\chi(\Omega)} = 2 + o(1)$ for even $N - M - 1$, strongly indicating a topology homeomorphic to the $S^{N-M-1}$ sphere. This regime is the only nontrivial one found with linear covariance $f(q) = q$, where the solution manifold must be a sphere if it is not empty.

**Regime II: $\overline{\chi(\Omega)}$ large and negative, isolated contributions at $m = \pm m_*$.** This regime is found when the magnitude of the target value $V_0$ is less than the onset $V_{\mathrm{on}}$, $\mathrm{Re}\,\mathcal{S}(0) > 0$, and the value of the action at $m = 0$ is complex. The dominant contribution to the average Euler characteristic comes from the equator at $m = 0$, but the complexity of the action implies that the Euler characteristic is negative. While the topology of the manifold is not necessarily connected in this regime, holes are more numerous than components. Since $V_0^2 < V_{\mathrm{on}}^2$, there are isolated contributions to $\overline{\chi(\Omega)}$ at $m = \pm m_*$. This implies a temperate band of relative simplicity: given a random point on the sphere, the nearest parts of the solution manifold are unlikely to have holes or disconnected components.

**Regime III: $\overline{\chi(\Omega)}$ large and negative, no contribution at $m = \pm m_*$.** The same as Regime II, but with $V_0^2 > V_{\mathrm{on}}^2$. The solutions at $m = \pm m_*$ no longer exist, and nontrivial contributions to the Euler characteristic are made all the way to the edges of the solution manifold.

---

[5]In the following we characterize regimes by values of $\overline{\chi(\Omega)}$. These should be understood as their values in *even* dimensions, since in odd dimensions the Euler characteristic is always identically zero. We do not expect the qualitative results to change depending on the evenness or oddness of the manifold dimension.

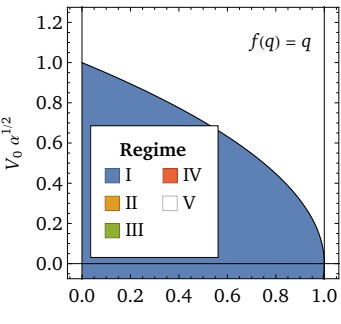 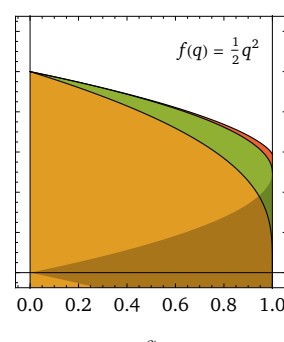 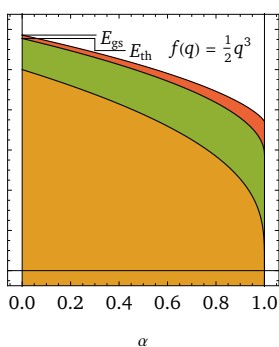

Figure 3: **Topological phase diagram.** Topological phases of the problem for three different homogeneous covariance functions. The regimes are defined in the text and depicted as cartoons in Fig. 2. The shaded region in the center panel shows where these results are unstable to RSB. In the limit of $\alpha \to 0$, the behavior of level sets of the spherical spin glasses are recovered: the righthand plot shows how the ground state energy $E_{gs}$ and threshold energy $E_{th}$ of the 3-spin spherical model correspond with the limits of the satisfiability and shattering transitions in the pure cubic problem. Note that for mixed models with inhomogeneous covariance functions, $E_{th}$ is not the lower limit of $V_{sh}$.

**Regime IV: $\overline{\chi(\Omega)}$ large and positive.** This regime is found when the magnitude of the target value $V_0$ is greater than the shattering value $V_{sh}$ and $\mathcal{S}(0) > 0$. Above the shattering transition the effective action is real everywhere, and its value at the equator is the dominant contribution. Large connected components of the manifold may or may not exist, but small disconnected components outnumber holes.[6]

**Regime V: $\overline{\chi(\Omega)}$ very small.** Here $\frac{1}{N} \log \overline{\chi(\Omega)} < 0$, indicating that the average Euler characteristic shrinks exponentially with $N$. Under most conditions we conclude this is the UNSAT regime where no manifold exists, but there may be circumstances where part of this regime is characterized by nonempty solution manifolds that are overwhelmingly likely to have Euler characteristic zero.

The distribution of these phases for situations with homogeneous polynomial constraint functions is shown in Fig. 3. For purely linear models, the only two regimes are I and V, separated by a satisfiability transition at $V_{SAT*}$. This is expected: the intersection of a plane and a sphere is another sphere, and therefore a model of linear constraints in a spherical configuration space can only produce a solution manifold consisting of a single sphere, or the empty set. For purely nonlinear models, regime I does not appear, while the other three nontrivial regimes do. Regimes II and III are separated by the onset transition at $V_{on}$, while III and IV are separated by the shattering transition at $V_{sh}$. Finally, IV and V are now separated by the satisfiability transition at $V_{SAT}$.

An interesting feature occurs in the limit of $\alpha$ to zero. If $V_0$ is likewise rescaled in the correct way, the limit of these phase boundaries approaches known landmark energy values in the pure spherical spin glasses. In particular, the limit $\alpha \to 0$ of the scaled satisfiability transition

---

[6]We interpret the large Euler characteristic to indicate a manifold with many (topologically) spherical disconnected components because the manifold is formed by the process of repeatedly taking non-self-intersecting slices of the previous manifold, starting with a sphere. Therefore, an outcome consisting mostly of (topological) spheres seems most plausible. However, a large Euler characteristic is also consistent with a variety of connected product manifolds, among other exotic possibilities. Definitely ruling out such scenarios is not within the scope of this paper.

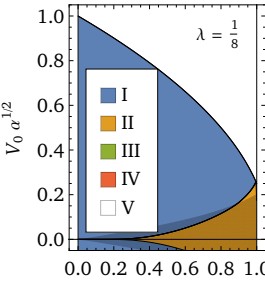
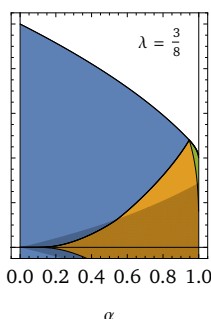
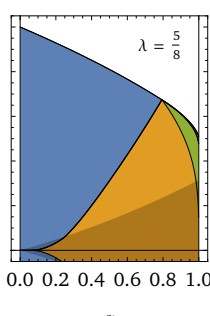
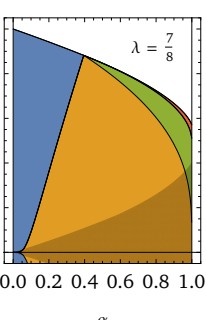

Figure 4: **Linear–quadratic crossover.** Topological phases for models with a covariance function $f(q) = (1-\lambda)q + \lambda \frac{1}{2}q^2$ for several values of $\lambda$, interpolating between homogeneous linear ($\lambda = 0$) and quadratic ($\lambda = 1$) constraints. The regimes are defined in the text and depicted as cartoons in Fig. 2. The shaded region on each plot shows where these results are unstable to RSB.

$V_{\text{SAT}}\sqrt{\alpha}$ approaches the ground state energy $E_{\text{gs}}$, while the limit $\alpha \to 0$ of the scaled shattering transition $V_{\text{sh}}\sqrt{\alpha}$ approaches the threshold energy $E_{\text{th}}$. The correspondence between ground state and satisfiability is expected: when the energy of a level set is greater in magnitude than the ground state, the level set will usually be empty. The correspondence between the threshold and shattering energies is also intuitive, since the threshold energy is typically understood as the point where the landscape fractures into pieces. However, this second correspondence is only true for the pure spherical models with homogeneous $f(q)$. For any other model with an inhomogeneous $f(q)$, $E_{\text{sh}}^2 < E_{\text{th}}^2$. This may have implications for dynamics in these mixed models, and we discuss them at length in Section 3.

Rich coexistence between all four regimes occurs in models with mixed linear and nonlinear constraints. Fig. 4 shows examples of the phase diagrams for models with a covariance function that interpolates between pure linear ($\lambda = 0$) and pure quadratic ($\lambda = 1$). A new phase boundary appears separating regimes I and II, defined as the point where the real part of the action at $m = 0$ changes from negative to positive. In purely quadratic case, and in mixed linear and nonlinear cases, there is a substantial region of the phase diagram shown in Appendix C.2 to be susceptible to RSB, especially for small $V_0$ and large $\alpha$. Future research into the structure of solutions in this regime is merited.

# 3 Implications for the dynamics of spherical spin glasses

When $M = 1$ the solution manifold corresponds to the energy level set of a spherical spin glass with energy density $E = V_0/\sqrt{N}$. All the results from the previous sections follow, and can be translated to the spin glasses by taking the limit $\alpha \to 0$ while keeping $E = V_0 \alpha^{1/2}$ fixed.[7] With a little algebra this procedure yields

$$E_{\text{on}} = \pm\sqrt{2f(1)}, \qquad E_{\text{sh}} = \pm\sqrt{4f(1)\left(1 - \frac{f(1)}{f'(1)}\right)}, \tag{19}$$

for the onset and shattering energies. The same limit taken for $V_{\text{SAT}}\alpha^{1/2}$ coincides with the ground state energy $E_{\text{gs}}$. In fact, for all energies below the threshold energy $E_{\text{th}}$ (where minima

---

[7]It is plausible that the limit of $N \to \infty$ implicit in the saddle point expansion and the limit of $\alpha \to 0$ taken here do not commute, and that $M = 1$ should be set from the beginning of the calculation. However, in this case the two procedures do commute. The $\alpha \to 0$ limit accomplishes only the elimination of the first term from the effective action (10), while following Appendix A with $M = 1$ from the outset results in the same term not appearing in the effective action because it is of subleading order in $N$.

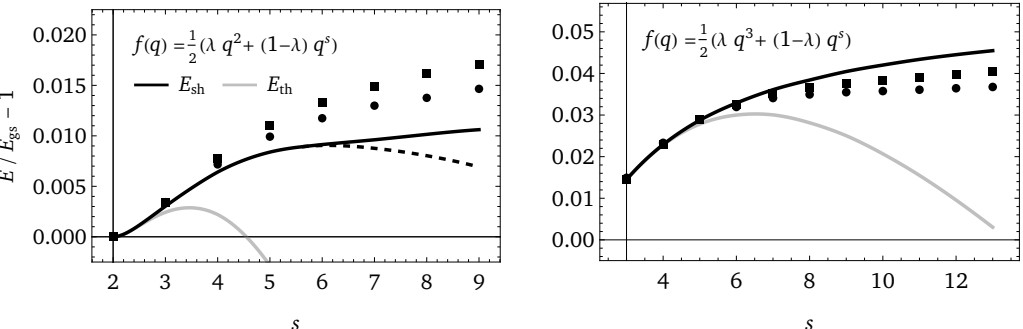

Figure 5: **Is the shattering energy a dynamic threshold?** Comparison of the shattering energy $E_{\text{sh}}$ with the asymptotic performance of gradient descent from a random initial condition in $p + s$ models with $p = 2$ and $p = 3$ and varying $s$. The values of $\lambda$ depend on $p$ and $s$ and are taken from [33]. The points show the asymptotic performance extrapolated using two different methods and have unknown uncertainty, from [33]. Also shown is the annealed threshold energy $E_{\text{th}}$, where marginal minima are the most common type of stationary point. The section of $E_{\text{sh}}$ that is dashed on the left plot indicates the continuation of the annealed result, whereas the solid portion gives the quenched prediction.

become more numerous than saddle points in the spin glass energy function) the logarithm of the average Euler characteristic is precisely the complexity of stationary points of the spin glass energy. In this regime, the Euler characteristic is dominated by contributions coming from the sphere-like slices of the energy basins directly above minima.

For the pure $p$-spin spherical spin glasses, which have homogeneous covariance functions $f(q) = \frac{1}{2}q^p$, the shattering energy is $E_{\text{sh}} = \sqrt{2(p-1)/p}$, precisely the same as the threshold energy $E_{\text{th}}$ [30]. This is intuitive, since threshold energy is widely understood as the place where level sets are broken into pieces. However, for general mixed models with inhomogeneous covariance functions the threshold energy is

$$E_{\text{th}} = \pm \frac{f(1)[f''(1) - f'(1)] + f'(1)^2}{f'(1)\sqrt{f''(1)}}, \tag{20}$$

which satisfies $|E_{\text{sh}}| \leq |E_{\text{th}}|$. Therefore, as one descends in energy one will generically meet the shattering energy before the threshold energy. This is perhaps unexpected, since one might imagine that where level sets of the energy break into many pieces would coincide with the largest concentration of shallow minima in the landscape. We see here that this isn't the case.

This fact mirrors another that was made clear recently: when gradient decent dynamics are run on these models, they will asymptotically reach an energy above the threshold energy [31–33]. The old belief that the threshold energy qualitatively coincides with a kind of shattering of the landscape is one source of the expectation that it should coincide with the dynamic limit. Motivated by our discovery that the actual shattering energy is different from the threshold energy, we make a comparison of it with existing data on asymptotic dynamics.

Measurements of the asymptotic energies reached by dynamics were recently taken in [33] for two different classes of models with inhomogeneous $f(q)$, with

$$f(q) = \frac{1}{2}\Big[\lambda q^p + (1-\lambda)q^s\Big]. \tag{21}$$

The authors of [33] studied models with this covariance for $p = 2$ and $p = 3$ while varying $s$. In both cases, the relative weight $\lambda$ between the two terms varies with $s$ and was chosen to maximize a heuristic to increase the chances of seeing nontrivial behavior. The authors

numerically integrated the dynamic mean field theory (DMFT) equations for gradient descent in these models from a random initial condition to large but finite time, then attempted to extrapolate the infinite-time behavior by two different methods. The black symbols in Fig. 5 show the measurements taken from [33]. The difference between the two extrapolations is not critical here, see the original paper for details. We simply note that the authors of [33] did not associate an uncertainty with them, nor were they confident that they are unbiased estimates of the asymptotic value.

Fig. 5 also shows the shattering and annealed threshold energies as a function of $s$. The solid lines come from using *Mathematica*'s `Interpolation` function to create a smooth function $\lambda(s)$ through the values used in [33]. For the 2+$s$ models for sufficiently large $s$, the ground state is described by a 1FRSB order and both the threshold energy and shattering energy calculated using an annealed average are likely inaccurate [34]. In Appendix D we calculate the quenched ground state and shattering energies for these models consistent with the 1FRSB equilibrium order. In the left panel of Fig. 5, the solid line shows the quenched calculation, while the dashed line shows the annealed formula (19).

Is the shattering energy consistent with the dynamic threshold for gradient descent from a random initial condition? The evidence in Fig. 5 is compelling but inconclusive. The difference between the shattering energy and the extrapolated DMFT data is about the same as the difference between the values predicted by the two extrapolation methods. If both extrapolation methods suffer from similar systematic biases, it is plausible the true value is the shattering energy. However, better estimates of the asymptotic values are needed to support or refute this conjecture. This motivates working to integrate the DMFT equations to longer times, or else look for analytic asymptotic solutions that approach $E_{\text{sh}}$.

The shattering energy appears consistent with the energy reached by gradient descent from a uniformly random initial condition, but other algorithms find minima at other energies. Optimal message passing algorithms were shown to find configurations at an energy level where another topological property—the overlap gap property—transitions, and this energy level is believed to bound from below all polynomial-time algorithms [35–37]. On the other hand, physically inspired modifications of gradient descent—notably, drawing the initial condition from a nonuniform distribution like the Boltzmann distribution with a finite temperature— find energy configurations with energies lower than those found with gradient descent from a uniform initial condition [31, 32]. If the topological transition described in this paper does predict the asymptotic performance of gradient descent from a uniform initial condition, then it provides a topological bound from above for the performance of reasonable algorithms that terminate in minima. It is unknown whether the performance of gradient descent from better initial conditions, or of other algorithms like simulated annealing, can be predicted with a topological property.

Finally, a common extension of the spherical spin glasses is to add a deterministic piece to the energy, sometimes called a signal or a spike. Recent work argued that gradient descent can avoid being trapped by the minima that typically trap dynamics and reach the vicinity of the signal if the set of typically trapping minima has been destabilized by the presence of the signal [38, 39]. The authors of Ref. [39] conjecture based on DMFT data for $2+3$ mixed spherical spin glasses that the typically trapping minima are those at the threshold energy $E_{\text{th}}$. However, as discussed above, Ref. [33] demonstrated that in signal-free mixed $p+s$ spherical spin glasses $E_{\text{th}}$ is not the energy of typical trapping minima, and furthermore that, when $p$ and $s$ are small, the difference between $E_{\text{th}}$ and the energy of the actual trapping minima is difficult to resolve with the current precision of DMFT integration schemes. Therefore, it is plausible that the picture described in Ref. [39] is correct except that the set of minima that must be destabilised to reach the signal is that at the typical trapping energy of the isotropic problem and not the threshold energy $E_{\text{th}}$. If the conjecture made in this paper is true, then this

typical trapping energy is the shattering energy $E_{sh}$. Comparing the predictions of Ref. [39] to DMFT simulations of a model with better separation between $p$ and $s$ would help resolve this question.

## 4 Conclusion

We have shown how to calculate the average Euler characteristic of the solution manifold in a simple model of random continuous constraint satisfaction. The results constrain the topology of this manifold, revealing when it is connected and trivial, when it is extensive but topologically nontrivial, and when it is shattered into disconnected pieces.

This calculation has novel implications for the geometry of the energy landscape in the spherical spin glasses, where it reveals a previously unknown landmark energy $E_{sh}$. This shattering energy is where the topological calculation implies that the level set of the energy breaks into disconnected pieces, and differs from the threshold energy $E_{th}$ in mixed models. It's possible that $E_{sh}$ is the asymptotic energy reached by gradient descent from a random initial condition in such models, but the quality of the currently available data makes this conjecture inconclusive.

Our work also highlights a limitation of using the statistics of stationary points of an energy or cost function to infer topological properties of the level sets. In the mixed spherical spin glasses, neither one nor two stationary point statistics reveal the presence of the topologically significant energy density $E_{sh}$ [31, 40, 41]. If the shattering energy is found conclusively to be the dynamic threshold for gradient descent this failure will be all the more serious. It may be enlightening to return to old problems of mean-field landscape analysis with this approach in hand, including in the analysis of the TAP free energy in many spin-glass settings [42–44].

This paper has focused on equality constraints, while most existing studies of constraint satisfaction study inequality constraints [2, 45–48]. To generalize the technique developed in this paper to such cases is not a trivial extension. The set of solutions to such problems are manifolds with boundary, and these boundaries are often not smooth. To study such cases with these techniques will require using extensions of the Morse theory for manifolds with boundary, and will be the subject of future work.

## Acknowledgments

The authors thank Pierfrancesco Urbani for helpful conversations on these topics, and Giampaolo Folena for supplying his DMFT data for the spherical spin glasses.

**Funding information** JK-D is supported by FAPESP Young Investigator Grant No. 2024/11114-1. JK-D also received support from the Simons Foundation Targeted Grant to ICTP-SAIFR and a DYNSYSMATH Specific Initiative of the INFN.

# A  Details of the calculation of the average Euler characteristic

Our starting point is the expression (8). To evaluate the average of $\chi$ over the random constraints, we first translate the $\delta$-function and determinant to integral form, with

$$\delta\big(\partial L(\mathbf{x},\boldsymbol{\omega})\big) = \int \frac{d\hat{\mathbf{x}}}{(2\pi)^N} \frac{d\hat{\boldsymbol{\omega}}}{(2\pi)^{M+1}} e^{i[\hat{\mathbf{x}},\hat{\boldsymbol{\omega}}]\cdot\partial L(\mathbf{x},\boldsymbol{\omega})}, \tag{A.1}$$

$$\det \partial\partial L(\mathbf{x},\boldsymbol{\omega}) = \int d\bar{\boldsymbol{\eta}}\, d\boldsymbol{\eta}\, d\bar{\boldsymbol{\gamma}}\, d\boldsymbol{\gamma}\, e^{-[\bar{\boldsymbol{\eta}},\bar{\boldsymbol{\gamma}}]^T \partial\partial L(\mathbf{x},\boldsymbol{\omega})[\boldsymbol{\eta},\boldsymbol{\gamma}]}, \tag{A.2}$$

where $\hat{\mathbf{x}}$ and $\hat{\boldsymbol{\omega}}$ are ordinary vectors and $\bar{\boldsymbol{\eta}}$, $\boldsymbol{\eta}$, $\bar{\boldsymbol{\gamma}}$, and $\boldsymbol{\gamma}$ are Grassmann vectors. With these expressions substituted into (8), the result is an integral over an exponential whose argument is linear in the random functions $V_k$.

To make the calculation compact, we introduce superspace coordinates [49]. An introduction to the use of superspace coordinates in mean field theoretical calculations, including definitions of operators like the superdeterminant using the same conventions as the present article, can be found in Appendix A of Ref. [23]. Introducing the Grassmann indices $\bar{\theta}_1$ and $\theta_1$, we define the supervectors

$$\boldsymbol{\phi}(1) = \mathbf{x} + \bar{\theta}_1 \boldsymbol{\eta} + \bar{\boldsymbol{\eta}}\theta_1 + \bar{\theta}_1\theta_1 i\hat{\mathbf{x}}, \qquad \sigma_k(1) = \omega_k + \bar{\theta}_1\gamma_k + \bar{\gamma}_k\theta_1 + \bar{\theta}_1\theta_1 i\hat{\omega}_k, \tag{A.3}$$

with associated measures

$$d\boldsymbol{\phi} = d\mathbf{x}\, \frac{d\hat{\mathbf{x}}}{(2\pi)^N}\, d\bar{\boldsymbol{\eta}}\, d\boldsymbol{\eta}, \qquad d\boldsymbol{\sigma} = d\boldsymbol{\omega}\, \frac{d\hat{\boldsymbol{\omega}}}{(2\pi)^{M+1}}\, d\bar{\boldsymbol{\gamma}}\, d\boldsymbol{\gamma}. \tag{A.4}$$

The Euler characteristic can be expressed using these supervectors as

$$\chi(\Omega) = \int d\boldsymbol{\phi}\, d\boldsymbol{\sigma}\, e^{\int d1\, L(\boldsymbol{\phi}(1),\boldsymbol{\sigma}(1))} \tag{A.5}$$

$$= \int d\boldsymbol{\phi}\, d\boldsymbol{\sigma}\, \exp\left\{\int d1\left[H\big(\boldsymbol{\phi}(1)\big) + \frac{1}{2}\sigma_0(1)\big(\|\boldsymbol{\phi}(1)\|^2 - N\big) + \sum_{k=1}^{M} \sigma_k(1)\big(V_k\big(\boldsymbol{\phi}(1)\big) - V_0\big)\right]\right\},$$

where $d1 = d\bar{\theta}_1\, d\theta_1$ is the integration measure over both Grassmann indices. Since this is an exponential integrand linear in the Gaussian functions $V_k$, we can take their average to find

$$\overline{\chi(\Omega)} = \int d\boldsymbol{\phi}\, d\boldsymbol{\sigma}\, \exp\left\{\int d1\left[H(\boldsymbol{\phi}(1)) + \frac{1}{2}\sigma_0(1)\big(\|\boldsymbol{\phi}(1)\|^2 - N\big) - V_0 \sum_{k=1}^{M}\sigma_k(1)\right]\right.$$
$$\left. + \frac{1}{2}\int d1\, d2 \sum_{k=1}^{M} \sigma_k(1)\sigma_k(2) f\left(\frac{\boldsymbol{\phi}(1)\cdot\boldsymbol{\phi}(2)}{N}\right)\right\}. \tag{A.6}$$

This is a super-Gaussian integral in the super-Lagrange multipliers $\sigma_k$ with $1 \le k \le M$. Performing that integral yields

$$\overline{\chi(\Omega)} = \int d\boldsymbol{\phi}\, d\sigma_0\, \exp\left\{\int d1\left[H(\boldsymbol{\phi}(1)) + \frac{1}{2}\sigma_0(1)\big(\|\boldsymbol{\phi}(1)\|^2 - N\big)\right]\right. \tag{A.7}$$
$$\left. - \frac{M}{2}V_0^2 \int d1\, d2\, f\left(\frac{\boldsymbol{\phi}(1)\cdot\boldsymbol{\phi}(2)}{N}\right)^{-1} - \frac{M}{2}\log\operatorname{sdet} f\left(\frac{\boldsymbol{\phi}(1)\cdot\boldsymbol{\phi}(2)}{N}\right)\right\}.$$

The supervector $\boldsymbol{\phi}$ enters this expression as a function only of the scalar product with itself and with the vector $\mathbf{x}_0$ inside the height function $H(\mathbf{x}) = \mathbf{x}_0 \cdot \mathbf{x}$. We therefore make a change of variables to the superoperator $\mathbb{Q}$ and the supervector $\mathbb{M}$ defined by

$$\mathbb{Q}(1,2) = \frac{\boldsymbol{\phi}(1)\cdot\boldsymbol{\phi}(2)}{N}, \qquad \mathbb{M}(1) = \frac{\boldsymbol{\phi}(1)\cdot\mathbf{x}_0}{N}. \tag{A.8}$$

These new variables can replace $\boldsymbol{\phi}$ in the integral using a generalized Hubbard–Stratonovich transformation, which yields

$$\overline{\chi(\Omega)} = \frac{1}{2} \int d\mathbb{Q}\, d\mathbb{M}\, d\sigma_0 \left( [\text{sdet}(\mathbb{Q} - \mathbb{M}\mathbb{M}^T)]^{\frac{1}{2}} + O(N^{-1}) \right) \exp \left\{ \frac{N}{2} \log \text{sdet}(\mathbb{Q} - \mathbb{M}\mathbb{M}^T) \right. \tag{A.9}$$

$$\left. + N \int d1 \left[ \mathbb{M}(1) + \frac{1}{2} \sigma_0(1) \big( \mathbb{Q}(1,1) - 1 \big) \right] - \frac{M}{2} V_0^2 \int d1\, d2\, f(\mathbb{Q})^{-1}(1,2) - \frac{M}{2} \log \text{sdet}\, f(\mathbb{Q}) \right\},$$

where we show the asymptotic value of the prefactor in Appendix B. To move on from this expression, we need to expand the superspace notation. We can write

$$\mathbb{Q}(1,2) = C - R(\bar{\theta}_1 \theta_1 + \bar{\theta}_2 \theta_2) - G(\bar{\theta}_1 \theta_2 + \bar{\theta}_2 \theta_1) - D\bar{\theta}_1 \theta_1 \bar{\theta}_2 \theta_2$$
$$+ (\bar{\theta}_1 + \bar{\theta}_2)H + \bar{H}(\theta_1 + \theta_2) - (\bar{\theta}_1 \theta_1 \bar{\theta}_2 + \bar{\theta}_2 \theta_2 \bar{\theta}_1)i\hat{H} - \bar{\hat{H}}(\theta_1 \bar{\theta}_2 \theta_2 + \theta_1 \bar{\theta}_1 \theta_1), \tag{A.10}$$

and

$$\mathbb{M}(1) = m + \bar{\theta}_1 H_0 + \bar{H}_0 \theta_1 + i\hat{m}\bar{\theta}_1 \theta_1, \tag{A.11}$$

with associated measures

$$d\mathbb{Q} = dC\, dR\, dG\, \frac{dD}{(2\pi)^2}\, d\bar{H}\, dH\, d\bar{\hat{H}}\, d\hat{H}, \qquad d\mathbb{M} = dm\, \frac{d\hat{m}}{2\pi}\, d\bar{H}_0\, dH_0. \tag{A.12}$$

The order parameters $C$, $R$, $G$, $D$, $m$, and $\hat{m}$ are ordinary numbers defined by

$$C = \frac{\mathbf{x} \cdot \mathbf{x}}{N}, \quad R = -i\frac{\mathbf{x} \cdot \hat{\mathbf{x}}}{N}, \quad G = \frac{\bar{\boldsymbol{\eta}} \cdot \boldsymbol{\eta}}{N}, \quad D = \frac{\hat{\mathbf{x}} \cdot \hat{\mathbf{x}}}{N}, \quad m = \frac{\mathbf{x}_0 \cdot \mathbf{x}}{N}, \quad \hat{m} = -i\frac{\mathbf{x}_0 \cdot \hat{\mathbf{x}}}{N}, \tag{A.13}$$

while $\bar{H}$, $H$, $\bar{\hat{H}}$, $\hat{H}$, $\bar{H}_0$ and $H_0$ are Grassmann numbers defined by

$$\bar{H} = \frac{\bar{\boldsymbol{\eta}} \cdot \mathbf{x}}{N}, \quad H = \frac{\boldsymbol{\eta} \cdot \mathbf{x}}{N}, \quad \bar{\hat{H}} = -i\frac{\bar{\boldsymbol{\eta}} \cdot \hat{\mathbf{x}}}{N}, \quad \hat{H} = -i\frac{\boldsymbol{\eta} \cdot \hat{\mathbf{x}}}{N}, \quad \bar{H}_0 = \frac{\bar{\boldsymbol{\eta}} \cdot \mathbf{x}_0}{N}, \quad H_0 = \frac{\boldsymbol{\eta} \cdot \mathbf{x}_0}{N}. \tag{A.14}$$

We can treat the integral over $\sigma_0$ immediately. It gives

$$\int d\sigma_0\, e^{N \int d1 \frac{1}{2} \sigma_0(1)(\mathbb{Q}(1,1)-1)} = 2 \times 2\pi \delta(C-1) \delta(G+R) \bar{H}H. \tag{A.15}$$

This therefore sets $C = 1$ and $G = -R$ in the remainder of the integrand, as well as removing all dependence on $\bar{H}$ and $H$. With these solutions inserted, the remaining terms in the exponential expand to give

$$\text{sdet}(\mathbb{Q} - \mathbb{M}\mathbb{M}^T) = 1 + \frac{(1-m^2)D + \hat{m}^2 - 2Rm\hat{m}}{R^2} - \frac{6}{R^4}\bar{H}_0 H_0 \bar{\hat{H}}\hat{H} \tag{A.16}$$

$$+ \frac{2}{R^3} \left[ (mR - \hat{m})(\bar{\hat{H}}H_0 + \bar{H}_0 \hat{H}) - (D + R^2)\bar{H}_0 H_0 + (1-m^2)\bar{\hat{H}}\hat{H} \right],$$

$$\text{sdet}\, f(\mathbb{Q}) = 1 + \frac{Df(1)}{R^2 f'(1)} + \frac{2f(1)}{R^3 f'(1)}\bar{\hat{H}}\hat{H}, \tag{A.17}$$

$$\int d1\, d2\, f(\mathbb{Q})^{-1}(1,2) = \frac{1}{f(1)} \left( 1 + \frac{R^2 f'(1)}{Df(1)} \right)^{-1} + 2\frac{Rf'(1)}{(Df(1) + R^2 f'(1))^2}\bar{\hat{H}}\hat{H}. \tag{A.18}$$

The Grassmann terms in these expressions do not contribute to the effective action, but will be important in our derivation of the prefactor for the exponential around the stationary points at $\pm m_*$. The substitution of these expressions into (A.9) without the Grassmann terms yields

$$\overline{\chi(\Omega)} = \left( \frac{N}{2\pi} \right)^2 \int dR\, dD\, dm\, d\hat{m}\, g(R, D, m, \hat{m})\, e^{N\mathcal{S}_\chi(R,D,m,\hat{m})}, \tag{A.19}$$

where $g$ is a prefactor of $o(N^0)$ detailed in the following appendix, $\mathcal{S}_\chi$ is an effective action defined by

$$
\begin{aligned}
\mathcal{S}_\chi(R, D, m, \hat{m}) = &-\hat{m} - \frac{\alpha}{2}\left[\log\left(1 + \frac{f'(1)D}{f'(1)R^2}\right) + \frac{V_0^2}{f(1)}\left(1 + \frac{f'(1)R^2}{f(1)D}\right)^{-1}\right] \\
&+ \frac{1}{2}\log\left(1 + \frac{(1-m^2)D + \hat{m}^2 - 2Rm\hat{m}}{R^2}\right),
\end{aligned}
\tag{A.20}
$$

and where we have introduced the ratio $\alpha = M/N$. The integral (A.19) can be evaluated to leading order in $N$ by a saddle point approximation. To get the formula (10) in the main text, we first extremize this expression with respect to $R$, $D$, and $\hat{m}$, which take the saddle-point values

$$
R = R_m, \qquad D = -\frac{m + R_m}{1 - m^2}R_m, \qquad \hat{m} = 0,
\tag{A.21}
$$

where $R_m$ is given by (11) from the main text.

# B Calculation of the prefactor of the average Euler characteristic

Because of our convention of including the appropriate factors of $2\pi$ in the superspace measure, super-Gaussian integrals do not produce such factors in our derivation. Prefactors to our calculation come from three sources: the introduction of $\delta$-functions to define the order parameters, integrals over Grassmann order parameters, and from the saddle point approximation to the large-$N$ integral. In addition, there are important contributions of a sign of the magnetization at the solution that arise from our super-Gaussian integrations.

## B.1 Contribution from the Hubbard–Stratonovich transformation

First, we examine the factors arising from the definition of order parameters. This begins by introducing to the integral (A.7) the factor of one

$$
1 = (2\pi)^3 \int d\mathbb{Q}\, d\mathbb{M}\, \delta\big(N\mathbb{Q}(1,2) - \boldsymbol{\phi}(1)\cdot\boldsymbol{\phi}(2)\big)\delta\big(N\mathbb{M}(1) - \mathbf{x}_0\cdot\boldsymbol{\phi}(1)\big),
\tag{B.1}
$$

where three factors of $2\pi$ come from the measures as defined in (A.12). Converting the $\delta$-function into an exponential integral yields

$$
\begin{aligned}
1 = \frac{1}{2}\int d\mathbb{Q}\, d\mathbb{M}\, d\tilde{\mathbb{Q}}\, d\tilde{\mathbb{M}}\, \exp\bigg\{&\frac{1}{2}\int d1\, d2\, \tilde{\mathbb{Q}}(1,2)\big(N\mathbb{Q}(1,2) - \boldsymbol{\phi}(1)\cdot\boldsymbol{\phi}(2)\big) \\
&+ \int d1\, i\tilde{\mathbb{M}}(1)\big(N\mathbb{M}(1) - \mathbf{x}_0\cdot\boldsymbol{\phi}(1)\big)\bigg\},
\end{aligned}
\tag{B.2}
$$

where the supervectors and measures for $\tilde{\mathbb{Q}}$ and $\tilde{\mathbb{M}}$ are defined analogously to those of $\mathbb{Q}$ and $\mathbb{M}$. This is now a super-Gaussian integral in $\boldsymbol{\phi}$, which can be performed to yield

$$
\begin{aligned}
\int d\boldsymbol{\phi}\, 1 = \frac{1}{2}\int d\mathbb{Q}\, d\mathbb{M}\, d\tilde{\mathbb{Q}}\, d\tilde{\mathbb{M}}\, \exp\bigg\{&\frac{N}{2}\int d1\, d2\, \tilde{\mathbb{Q}}(1,2)\mathbb{Q}(1,2) + N\int d1\, i\tilde{\mathbb{M}}(1)\mathbb{M}(1) \\
&- \frac{N}{2}\log\operatorname{sdet}\tilde{\mathbb{Q}} - \frac{N}{2}\int d1\, d2\, \tilde{\mathbb{M}}(1)\tilde{\mathbb{Q}}^{-1}(1,2)\tilde{\mathbb{M}}(2)\bigg\}.
\end{aligned}
\tag{B.3}
$$

We can perform the remaining super-Gaussian integral in $\tilde{\mathbb{M}}$ to find

$$
\int d\boldsymbol{\phi}\, 1 = \frac{1}{2} \int d\mathbb{Q}\, d\mathbb{M}\, d\tilde{\mathbb{Q}}\, (\mathrm{sdet}\, \tilde{\mathbb{Q}}^{-1})^{-\frac{1}{2}}
$$
$$
\times \exp\left\{ -\frac{N}{2} \log \mathrm{sdet}\, \tilde{\mathbb{Q}} + \frac{N}{2} \int d1\, d2\, \tilde{\mathbb{Q}}(1,2)\big[ \mathbb{Q}(1,2) - \mathbb{M}(1)\mathbb{M}(2) \big] \right\}.
$$
(B.4)

The integral over $\tilde{\mathbb{Q}}$ can be evaluated to leading order using the saddle point method. The integrand is stationary at $\tilde{\mathbb{Q}} = (\mathbb{Q} - \mathbb{M}\mathbb{M}^T)^{-1}$, and substituting this into the above expression results in the term $\frac{1}{2} \log \det(\mathbb{Q} - \mathbb{M}\mathbb{M}^T)$ in the effective action from (A.9). The saddle point also yields a prefactor of the form

$$
\left( \mathrm{sdet}_{\{1,2\},\{3,4\}} \frac{\partial^2 \frac{1}{2} \log \mathrm{sdet}\, \tilde{\mathbb{Q}}}{\partial \tilde{\mathbb{Q}}(1,2)\partial \tilde{\mathbb{Q}}(3,4)} \right)^{-\frac{1}{2}} = \left( \mathrm{sdet}_{\{1,2\},\{3,4\}} \tilde{\mathbb{Q}}^{-1}(3,1)\tilde{\mathbb{Q}}^{-1}(2,4) \right)^{-\frac{1}{2}} = 1, \quad \text{(B.5)}
$$

where the final superdeterminant is identically 1 for any superoperator $\tilde{\mathbb{Q}}$, not just its saddle-point value.[8] The Hubbard–Stratonovich transformation therefore contributes a factor of

$$
\frac{1}{2} \mathrm{sdet}(\mathbb{Q} - \mathbb{M}\mathbb{M}^T)^{\frac{1}{2}} = \frac{1}{2}\big[ (C - m^2)(D + \hat{m}^2) + (R - m\hat{m})^2 \big]^{\frac{1}{2}} G^{-1},
$$
(B.6)

to the prefactor at the largest order in $N$.

## B.2 Sign of the prefactor

The superspace notation papers over some analytic differences between branches of the logarithm that are not important for determining the saddle point but are important to getting correctly the sign of the prefactor. For instance, consider the superdeterminant of $\mathbb{Q}$ from (A.10) (dropping the fermionic order parameters for a moment for brevity),

$$
\mathrm{sdet}\, \mathbb{Q} = \frac{CD + R^2}{G^2}.
$$
(B.7)

The numerator and denominator arise from the determinant in the sector of ordinary number and Grassmann number basis elements for the superoperator, respectively. In our calculation, such superdeterminants appear after Gaussian integrals, like

$$
\int d\boldsymbol{\phi}\, \exp\left\{ -\frac{1}{2} \int d1\, d2\, \boldsymbol{\phi}(1)\mathbb{Q}(1,2)\boldsymbol{\phi}(2) \right\} = (\mathrm{sdet}\, \mathbb{Q})^{-\frac{1}{2}} = (CD + R^2)^{-\frac{1}{2}} G.
$$
(B.8)

Here we emphasize that in the expanded result of the integral, the factor from the denominator of the superdeterminant enters not as $(G^2)^{\frac{1}{2}} = |G|$ but as $G$, including its sign. Therefore, when we write in the effective action $-\frac{1}{2} \log \mathrm{sdet}\, \mathbb{Q}$, we should really be writing

$$
\int d\boldsymbol{\phi}\, \exp\left\{ -\frac{1}{2} \int d1\, d2\, \boldsymbol{\phi}(1)\mathbb{Q}(1,2)\boldsymbol{\phi}(2) \right\} = \mathrm{sign}(G) e^{-\frac{1}{2} \log \mathrm{sdet}\, \mathbb{Q}}.
$$
(B.9)

In our calculation in Appendix A we elide this several times, and accumulate $M$ factors of $\mathrm{sign}(-Gf'(C)) = \mathrm{sign}(-G)$ from the Gaussian integral over Lagrange multipliers and $N$ factors of $\mathrm{sign}(-G)$ from the Hubbard–Stratonovich transformation. Since at all saddle points $G = -R$, we have

$$
\mathrm{sign}(R)^{N+M} e^{N\mathcal{S}_\chi(\tilde{\mathbb{Q}}, \mathbb{Q}, \mathbb{M})}.
$$
(B.10)

---

[8]The subscript notation in (B.5) indicates which superindices of the four-index superoperator associated with the Hessian belong to the domain and codomain, analogous to writing $\det A = \det_{ij} A_{ij}$ for a two-index complex-valued operator. In this case, the domain is indexed by $\{3,4\}$ and the codomain is indexed by $\{1,2\}$.

### B.3 Contribution from integrating the Grassmann order parameters

After integrating out the Lagrange multiplier enforcing the spherical constraint in (A.15), the Grassmann variables $\bar{H}$ and $H$ are eliminated from the integrand. This leaves dependence on $\hat{\bar{H}}$, $\hat{H}$, $\bar{H}_0$, and $H_0$. Expanding the contributions from (A.16), (A.17), and (A.18), the total contribution to the action is given by

$$\int d\hat{\bar{H}}\, d\hat{H}\, d\bar{H}_0\, dH_0 \exp\left\{ N \begin{bmatrix} \hat{\bar{H}} & \bar{H}_0 \end{bmatrix} \begin{bmatrix} h_1 & h_2 \\ h_2 & h_3 \end{bmatrix} \begin{bmatrix} \hat{H} \\ H_0 \end{bmatrix} + N h_4 \hat{\bar{H}} \hat{H} \bar{H}_0 H_0 \right\} = N^2(h_1 h_3 - h_2^2) + N h_4, \quad \text{(B.11)}$$

where

$$h_1 = \frac{1}{R}\left( \frac{1-m^2}{D(1-m^2)+R^2-2Rm\hat{m}+\hat{m}^2} - \alpha \frac{Df(1)^2 + R^2 f'(1)[V_0^2 + f(1)]}{[Df(1)+R^2 f'(1)]^2} \right), \quad \text{(B.12)}$$

$$h_2 = \frac{1}{R}\frac{Rm - \hat{m}}{D(1-m^2)+R^2-2Rm\hat{m}+\hat{m}^2}, \quad \text{(B.13)}$$

$$h_3 = -\frac{1}{R}\frac{D + R^2}{D(1-m^2)+R^2-2Rm\hat{m}+\hat{m}^2}, \quad \text{(B.14)}$$

$$h_4 = -\frac{1}{R^2}\frac{1}{D(1-m^2)+R^2-2Rm\hat{m}+\hat{m}^2}. \quad \text{(B.15)}$$

The contribution to the prefactor at leading order in $N$ is therefore

$$\frac{N^2}{R^2[D(1-m^2)+R^2-2Rm\hat{m}+\hat{m}^2]}\left( \alpha \frac{(D+R^2)[Df(1)^2 + R^2 f'(1)[V_0^2 + f(1)]]}{[Df(1)+R^2 f'(1)]^2} - 1 \right). \quad \text{(B.16)}$$

### B.4 Contribution from the saddle point approximation

We now want to evaluate the prefactor for the asymptotic value of $\overline{\chi(\Omega)}$. From the previous sections, the definition of the measures $d\mathbb{Q}$ and $d\mathbb{M}$ in (A.12), and the integral over $\sigma_0$ of (A.15), we can now see that the function $g(R, D, m, \hat{m})$ of (A.19) is given by

$$
\begin{aligned}
g(R, D, m, \hat{m}) = &-\frac{\text{sign}(R)^{N+M}}{R^3[D(1-m^2)+R^2-2Rm\hat{m}+\hat{m}^2]^{\frac{1}{2}}} \\
&\times \left( \alpha \frac{(D+R^2)[Df(1)^2 + R^2 f'(1)[V_0^2 + f(1)]]}{[Df(1)+R^2 f'(1)]^2} - 1 \right).
\end{aligned}
\quad \text{(B.17)}
$$

In regime I, there are two saddle points of the integrand that contribute to the asymptotic value of the integral, at $m = \pm m_*$ with $R = -m_*$, $D = 0$, and $\hat{m} = 0$. At this saddle point $\mathcal{S}_\chi = 0$. We can therefore write

$$\overline{\chi(\Omega)} = \sum_{m=\pm m_*} g(-m, 0, m, 0)\left[ \det \partial\partial \mathcal{S}_\chi(-m, 0, m, 0) \right]^{-\frac{1}{2}}, \quad \text{(B.18)}$$

where here $\partial = \left[\frac{\partial}{\partial R}, \frac{\partial}{\partial D}, \frac{\partial}{\partial m}, \frac{\partial}{\partial \hat{m}}\right]$ is the vector of derivatives with respect to the remaining order parameters. For both of the two saddle points, the determinant of the Hessian of the effective action evaluates to

$$\det \partial\partial \mathcal{S}_\chi = \left[ \frac{1}{(m_*)^4}\left( 1 - \frac{\alpha[V_0^2 + f(1)]}{f'(1)} \right) \right]^2, \quad \text{(B.19)}$$

whereas

$$g(\mp m_*, 0, \pm m_*, 0) = \frac{(\mp 1)^{N+M+1}}{(m_*)^4}\left( 1 - \frac{\alpha[V_0^2 + f(1)]}{f'(1)} \right). \quad \text{(B.20)}$$

The saddle point at $m = -m_*$, characterized by minima of the height function, always contributes with a positive term. On the other hand, the saddle point with $m = +m_*$, characterized by maxima of the height function, contributes with a sign depending on if $N + M + 1$ is even or odd. This follows from the fact that minima, with an index of $0$, have a positive contribution to the sum over stationary points, while maxima, with an index of $N - M - 1$, have a contribution that depends on the dimension of the manifold.

We have finally that, in regime I,

$$\overline{\chi(\Omega)} = 1 + (-1)^{N+M+1} + O(N^{-1}). \tag{B.21}$$

When $N + M + 1$ is odd, this evaluates to zero. In fact it must be zero to all orders in $N$, since for odd-dimensional manifolds the Euler characteristic is always zero. When $N + M + 1$ is even, we have $\overline{\chi(\Omega)} = 2$ to leading order in $N$, as specified in the main text.

## C  The average squared Euler characteristic

### C.1  Derivation

Here we calculate $\overline{\chi(\Omega)^2}$, the average of the squared Euler characteristic. This is accomplished by taking two copies of the integral (A.5), with

$$\chi(\Omega)^2 = \int d\boldsymbol{\phi}_1 \, d\boldsymbol{\sigma}_1 \, d\boldsymbol{\phi}_2 \, d\boldsymbol{\sigma}_2 \, e^{\int d1 \, [L(\boldsymbol{\phi}_1(1), \boldsymbol{\sigma}_1(1)) + L(\boldsymbol{\phi}_2(1), \boldsymbol{\sigma}_2(1))]}. \tag{C.1}$$

The same steps as in the derivation of the Euler characteristic follow. The result is the same as (A.9), but with the substitutions of the order parameters with matrices of order parameters,

$$\mathbb{Q} \mapsto \begin{bmatrix} \mathbb{Q}_{11} & \mathbb{Q}_{12} \\ \mathbb{Q}_{21} & \mathbb{Q}_{22} \end{bmatrix}, \qquad \mathbb{M} \mapsto \begin{bmatrix} \mathbb{M}_1 \\ \mathbb{M}_2 \end{bmatrix}, \tag{C.2}$$

where we have defined

$$\mathbb{Q}_{ij}(1,2) = \frac{1}{N} \boldsymbol{\phi}_i(1) \cdot \boldsymbol{\phi}_j(2), \qquad \mathbb{M}_i(1) = \frac{1}{N} \boldsymbol{\phi}_i(1) \cdot \mathbf{x}_0. \tag{C.3}$$

Expanding the superindices and applying the Dirac $\delta$-functions implied by the Lagrange multipliers associated with the spherical constraint (which set $C_{11} = C_{22} = 1$ and $G_{11} = -R_{11}$, $G_{22} = -R_{22}$), we arrive at an expression

$$\overline{\chi(\Omega)^2} \simeq \int dC_{12} \, dR_{11} \, dR_{12} \, dR_{21} \, dR_{22} \, dD_{11} \, dD_{12} \, dD_{22} \, dG_{12} \, dG_{21} \, dm_1 \, dm_2 \, d\hat{m}_1 \, d\hat{m}_2 \, e^{N\mathcal{S}_{\chi^2}}, \tag{C.4}$$

with another effective action defined by

$$\mathcal{S}_{\chi^2} = -\hat{m}_1 - \hat{m}_2 - \frac{\alpha}{2} \log \frac{\det A_1}{\det A_2} - \frac{\alpha V_0^2}{2} \begin{bmatrix} 0 & 1 & 0 & 1 \end{bmatrix} A_1^{-1} \begin{bmatrix} 0 \\ 1 \\ 0 \\ 1 \end{bmatrix} + \frac{1}{2} \log \frac{\det A_3}{\det A_4}, \tag{C.5}$$

with the matrices $A_1$, $A_2$, $A_3$, and $A_4$ given by

$$A_1 = \begin{bmatrix} D_{11}f'(1) & iR_{11}f'(1) & D_{12}f'(C_{12})+\Delta_{12}f''(C_{12}) & iR_{21}f'(C_{12}) \\ iR_{11}f'(1) & f(1) & iR_{12}f'(C_{12}) & f(C_{12}) \\ D_{12}f'(C_{12})+\Delta_{12}f''(C_{12}) & iR_{12}f'(C_{12}) & D_{22} & iR_{22}f'(1) \\ iR_{21}f'(C_{12}) & f(C_{12}) & iR_{22}f'(1) & f(1) \end{bmatrix}, \quad (C.6)$$

$$A_2 = \begin{bmatrix} 0 & R_{11}f'(1) & 0 & -G_{21}f'(C_{12}) \\ -R_{11}f'(1) & 0 & G_{12}f'(C_{12}) & 0 \\ 0 & -G_{12}f'(C_{12}) & 0 & R_{22}f'(1) \\ G_{21}f'(C_{12}) & 0 & -R_{22}f'(1) & 0 \end{bmatrix}, \quad (C.7)$$

$$A_3 = \begin{bmatrix} 1-m_1^2 & i(R_{11}-m_1\hat{m}_1) & C_{12}-m_1m_2 & i(R_{21}-m_1\hat{m}_2) \\ i(R_{11}-m_1\hat{m}_1) & D_{11}+\hat{m}_1^2 & i(R_{12}-m_2\hat{m}_1) & D_{12}+\hat{m}_1\hat{m}_2 \\ C_{12}-m_1m_2 & i(R_{12}-m_2\hat{m}_1) & 1-m_2^2 & i(R_{22}-m_2\hat{m}_2) \\ i(R_{21}-m_1\hat{m}_2) & D_{12}+\hat{m}_1\hat{m}_2 & i(R_{22}-m_2\hat{m}_2) & D_{22}+\hat{m}_2^2 \end{bmatrix}, \quad (C.8)$$

$$A_4 = \begin{bmatrix} 0 & R_{11} & 0 & -G_{21} \\ -R_{11} & 0 & G_{12} & 0 \\ 0 & -G_{12} & 0 & R_{22} \\ G_{21} & 0 & -R_{22} & 0 \end{bmatrix}, \quad (C.9)$$

and where $\Delta_{12} = G_{12}G_{21}-R_{12}R_{21}$. The effective action must be extremized over all the order parameters. We look for solutions in two regimes that are commensurate with the solutions found for the Euler characteristic. These correspond to $m_1 = m_2 = 0$ and $C_{12} = 0$, and $m_1 = m_2 = \pm m_*$ and $C_{12} = 1$. We restrict ourselves to cases with $f(0) = 0$, which correspond to constraint equations without a random constant term. We find such solutions, and in all cases they have

$$G_{12} = G_{21} = R_{12} = R_{21} = D_{12} = \hat{m}_1 = \hat{m}_2 = 0, \quad (C.10)$$

$$D_{ii} = -\frac{m+R_{ii}}{1-m^2}R_{ii}, \qquad R_{22} = R_{11}^{\dagger}, \qquad R_{11} = R_m, \quad (C.11)$$

where $\dagger$ denotes the complex conjugate and $R_m$ is the saddle point solution of (11). Upon substituting these solutions into the expressions above, we find in both cases that

$$\mathcal{S}_{\chi^2} = 2\operatorname{Re}\mathcal{S}_\chi, \quad (C.12)$$

as referenced in the main text. This corresponds with $\overline{\chi(\Omega)^2} \simeq [\overline{\chi(\Omega)}]^2$, justifying the 'annealed' approach we have taken in the rest of the paper.

## C.2 Instability to replica symmetry breaking

However, these solutions are not always the correct saddle point for evaluating the average squared Euler characteristic. When another solution is dominant, the dissonance between the average square and squared average indicates the necessity of a quenched calculation to determine the behavior of typical samples, and also indicates a likely instability to RSB. We can find these points of instability by examining the Hessian of the action of the average square of the Euler characteristic at $m = 0$. The stability of this matrix is not sufficient to determine if our solution is stable, since the many $\delta$-functions employed in our derivation ensure that the resulting saddle point is never at a true maximum with respect to some combinations of variables. We rather look for places where the stability of this matrix changes, indicating another solution branching from the existing one. However, we must neglect the branching of trivial solutions, which occur when $R_m$ goes from real- to complex-valued.

By examination of the results, it appears that nontrivial RSB instabilities occur along eigenvectors of the Hessian of $\mathcal{S}_{\chi^2}$ constrained to the subspace spanned by $C_{12}$, $R_{12}$, $R_{21}$, and $D_{12}$.

This may not be surprising, since these are the parameters that represent nontrivial correlations between the two copies of the system. We can therefore find the RSB instability by looking for nontrivial zeros of

$$\det \partial \partial \mathcal{S}_{\chi^2} \equiv \det \frac{\partial^2 \mathcal{S}_{\chi^2}}{\partial [C_{12}, R_{12}, R_{21}, D_{12}]^2}, \tag{C.13}$$

evaluated at the $m = 0$ solution described above. The resulting expression is usually quite heinous and we will not reproduce it in its general form in the text, but there is a regime where a dramatic simplification is possible. The instability always occurs along the direction $R_{21} = R_{12}^\dagger$, but when $R_m$ is real, $R_{11} = R_{22}$ and the instability occurs along the direction $R_{21} = R_{12}$. This allows us to examine a simpler action, and we find the determinant is proportional to two nontrivial factors, with

$$\det \partial \partial \mathcal{S}_{\chi^2} = -\frac{2B_1 B_2}{[r_* f'(1)]^3 [(1 + r_*) f(1) - r_* f'(1)]^7}. \tag{C.14}$$

If we define $r_* \equiv \lim_{m \to 0} R_m/m$, then the factors $B_1$ and $B_2$ are

$$B_1 = [(1 + r_*) f(1)]^3 - 3r_* [(1 + r_*) f(1)]^2 f'(1) + \alpha V_0^2 [2(1 + r_*) f'(0)^2 + r_* f'(1) f''(0)]$$
$$+ \alpha r_* f'(0)^2 f'(1) - [r_* f'(1)]^3 - (1 + r_*) f(1) \big(\alpha [f'(0)^2 + V_0^2 f''(0)] - 3[r_* f'(1)]^2\big), \tag{C.15}$$

$$B_2 = \big[(1 + r_*) f(1) - r_* f'(1)\big]^3 [f'(1)^2 - \alpha f'(0)^2] \big(f'(1)[(1 + r_*) f(1) - r_* f'(1)] - \alpha f'(0)^2\big)$$
$$- [\alpha V_0^2 r_* f'(1)]^2 f''(0) \big[(1 + r_*) f'(0)^2 + r_* f'(1) f''(0)\big]$$
$$- \alpha V_0^2 \big[(1 + r_*) f(1) - r_* f'(1)\big]^2 \Big[(1 + r_*) f'(0)^2 [\alpha f'(0)^2 - f'(1)^2]$$
$$+ r_* f'(1) f''(0) \Big(\alpha f'(0)^2 \frac{(1 + r_*) f(1) - 2r_* f'(1)}{(1 + r_*) f(1) - r_* f'(1)} - (1 - r_*) f'(1)^2\Big)\Big]. \tag{C.16}$$

As $\alpha$ is increased from zero, the first of these factors to go through zero represents the instability point. These formulas are responsible for defining the boundaries of the shaded regions in Fig. 3 and Fig. 4.

Surprisingly, this approach sees no signal of the replica symmetry breaking (RSB) transition previously found in [4]. The instability is predicted to occur when

$$V_0^2 > V_{\text{RSB}}^2 \equiv \frac{[f(1) - f(0)]^2}{\alpha f''(0)} - f(0) - \frac{f'(0)}{f''(0)}. \tag{C.17}$$

We conjecture that the RSB instability found in [4] is a trait of the cost function (3), and is not inherent to the structure of the solution manifold. Perhaps the best evidence for this is to consider the limit of $M = 1$, or $\alpha \to 0$ with $E = V_0 \sqrt{\alpha}$ held fixed, where this problem reduces to the level sets of the spherical spin glasses. The instability (C.17) implies for the pure spherical 2-spin model with $f(q) = \frac{1}{2} q^2$ that $E_{\text{RSB}} = \frac{1}{2}$, though nothing of note is known to occur in the level sets of 2-spin model at such an energy.

## D The quenched shattering energy

Here we share how the quenched shattering energy is calculated under a 1FRSB ansatz. To best make contact with prior work on the spherical spin glasses, we start with (A.6). The formula in a quenched calculation is almost the same as that for the annealed, but the order parameters $C$, $R$, $D$, and $G$ must be understood as $n \times n$ matrices rather than scalars. In principle $m$, $\hat{m}$, $\omega_0$, $\hat{\omega}_0$, $\omega_1$, and $\hat{\omega}_1$ should be considered $n$-dimensional vectors, but since in our ansatz replica

vectors are constant we can take them to be constant from the start. Expanding the superspace notation, setting $V_0 = E\sqrt{N/M}$, and taking $M = 1$, we have

$$
\begin{aligned}
\overline{\log \chi(\Omega)} = \lim_{n \to 0} \frac{\partial}{\partial n} \int & dC\, dR\, dD\, dG\, dm\, d\hat{m}\, d\omega_0\, d\hat{\omega}_0\, d\omega_1\, d\hat{\omega}_1 \exp N \left\{ n\hat{m} + \frac{i}{2}\hat{\omega}_0 \operatorname{Tr}(C - I) \right. \\
& -\omega_0 \operatorname{Tr}(G + R) - in\hat{\omega}_1 E + \frac{1}{2}\log\det\begin{bmatrix} C - m^2 & i(R - m\hat{m}) \\ i(R - m\hat{m}) & D - \hat{m}^2 \end{bmatrix} - \frac{1}{2}\log G^2 \\
& \left. -\frac{1}{2}\sum_{ab}^{n}\left[\hat{\omega}_1^2 f(C_{ab}) + (2i\omega_1\hat{\omega}_1 R_{ab} + \omega_1^2 D_{ab})f'(C_{ab}) + \omega_1^2(G_{ab}^2 - R_{ab}^2)f''(C_{ab})\right]\right\}.
\end{aligned} \tag{D.1}
$$

We now make a series of simplifications. Ward identities associated with the BRST symmetry possessed by the original action [50–52] indicate that

$$
\omega_1 D = -i\hat{\omega}_1 R, \qquad G = -R, \qquad \hat{m} = 0. \tag{D.2}
$$

Moreover, this problem with $m = 0$ has a close resemblance to the complexity of the spherical spin glasses. In both, at the BRST-symmetric saddle point the matrix $R$ is diagonal with $R = r_d I$ [53]. To investigate the shattering energy, we can restrict to solutions with $m = 0$ and look for the place where such solutions become complex. Inserting these simplifications, we have up to highest order in $N$

$$
\begin{aligned}
\overline{\log \chi(\Omega)} = \lim_{n \to 0} \frac{\partial}{\partial n} \int & dC\, dr_d\, d\hat{\omega}_0\, d\hat{\omega}_1 \exp N \left\{ \frac{i}{2}\hat{\omega}_0 \operatorname{Tr}(C - I) - in\hat{\omega}_1 E \right. \\
& \left. -i\frac{1}{2}n\omega_1^* \hat{\omega}_1 r_d f'(1) - \frac{1}{2}\sum_{ab}^{n}\hat{\omega}_1^2 f(C_{ab}) + \frac{1}{2}\log\det\left(\frac{-i\hat{\omega}_1}{\omega_1^* r_d}C + I\right)\right\},
\end{aligned} \tag{D.3}
$$

where $\omega_1^*$ is a constant set by satisfying the extremal equations for $D$. If we redefine $\hat{\beta} = -i\hat{\omega}_1$ and $\tilde{r}_d = \omega_1^* r_d$, we find

$$
\begin{aligned}
\overline{\log \chi(\Omega)} = \lim_{n \to 0} \frac{\partial}{\partial n} \int & dC\, d\hat{\beta}\, d\tilde{r}_d\, \hat{\omega}_0 \exp N \left\{ \frac{i}{2}\hat{\omega}_0 \operatorname{Tr}(C - I) + n\hat{\beta} E \right. \\
& \left. + n\frac{1}{2}\hat{\beta}\tilde{r}_d f'(1) + \frac{1}{2}\sum_{ab}^{n}\hat{\beta}^2 f(C_{ab}) + \frac{1}{2}\log\det\left(\frac{\hat{\beta}}{\tilde{r}_d}C + I\right)\right\},
\end{aligned} \tag{D.4}
$$

which is exactly the effective action for the supersymmetric complexity in the spherical spin glasses when in the regime where minima dominate [53]. As the effective action for the Euler characteristic, this expression is always valid. Following the same steps as in [53], we can write the continuum version of this action for arbitrary RSB structure in the matrix $C$ as

$$
\frac{1}{N}\overline{\log \chi(\Omega)} = \hat{\beta} E + \frac{1}{2}\hat{\beta}\tilde{r}_d f'(1) + \frac{1}{2}\int_0^1 dq \left[\hat{\beta}^2 f''(q)\chi(q) + \frac{1}{\chi(q) + \tilde{r}_d \hat{\beta}^{-1}}\right], \tag{D.5}
$$

where $\chi(q) = \int_1^q dq' \int_0^{q'} dq'' P(q'')$ and $P(q)$ is the distribution of off-diagonal elements of the matrix $C$ [54–56]. This action must be extremized over the function $\chi$ and the variables $\hat{\beta}$ and $\tilde{r}_d$, under the constraint that $\chi(q)$ is continuous, and that it has $\chi'(1) = -1$ and $\chi(1) = 0$, necessary for $P$ to be a well-defined probability distribution.

Now the specific form of replica symmetry breaking we expect to see is important. We want to study the mixed $2 + s$ models in the regime where they may have 1-full RSB in equilibrium [34]. For the Euler characteristic like the complexity, this will correspond to full RSB, in an

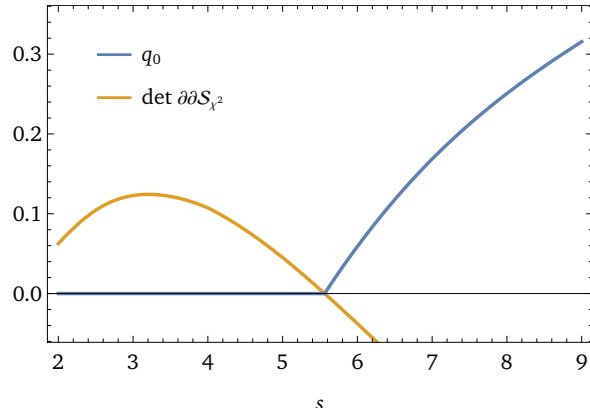

Figure 6: **Self-consistency between RSB instabilities.** Comparison between the predicted value $q_0$ for the FRSB solution at the shattering energy in $2+s$ models and the value of the determinant (C.14) used in the previous appendix to predict the point of RSB instability. The value of $s$ at which $q_0$ becomes nonzero is precisely the point where the determinant has a nontrivial zero.

analogous way that 1RSB equilibria give a RS complexity. Such order is characterized by a piecewise smooth $\chi$ of the form

$$\chi(q) = \begin{cases} \chi_0(q), & q < q_0, \\ 1-q, & q \geq q_0, \end{cases} \tag{D.6}$$

where

$$\chi_0(q) = \frac{1}{\hat{\beta}}[f''(q)^{-1/2} - \tilde{r}_d], \tag{D.7}$$

is the function implied by extremizing (D.5) over $\chi$ ignoring the continuity and other constraints. The variable $q_0$ must be chosen so that $\chi$ is continuous. The key difference between FRSB and 1FRSB in this setting is that in the former case the ground state has $q_0 = 1$, while in the latter the ground state has $q_0 < 1$.

We use this action to find the shattering energy in the following way. First, we know that the ground state energy is the place where the manifold and therefore the average Euler characteristic vanishes. Therefore, setting $\overline{\log \chi(\Omega)} = 0$ and solving for $E$ yields a formula for the ground state energy

$$E_{\mathrm{gs}} = -\frac{1}{\hat{\beta}}\left\{\frac{1}{2}\hat{\beta}\tilde{r}_d f'(1) + \frac{1}{2}\int_0^1 dq\left[\hat{\beta}^2 f''(q)\chi(q) + \frac{1}{\chi(q) + \tilde{r}_d\hat{\beta}^{-1}}\right]\right\}. \tag{D.8}$$

This expression can be maximized over $\hat{\beta}$ and $\tilde{r}_d$ to find the correct parameters at the ground state for a particular model. Then, the shattering energy is found by slowly lowering $q_0$ and solving the combined extremal and continuity problem for $\hat{\beta}$, $\tilde{r}_d$, and $E$ until $E$ reaches a maximum value and starts to decrease. This maximum is the shattering energy, since it is the point where the $m = 0$ solution becomes complex. Starting from this point, we take small steps in $s$ and $\lambda_s$, simultaneously extremizing, ensuring continuity, and maximizing $E$. This draws out the shattering energy across the entire range of $s$ plotted in Fig. 5. The transition to the RS solution occurs when the value $q_0$ that maximizes $E$ hits zero. We find that the transition between RS and FRSB is precisely predicted by the RSB instability calculated in Appendix C, as shown in Fig. 6.

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
