# Peer review of "On the topology of solutions to random continuous constraint satisfaction problems"

_SciPost Physics, doi:SciPost Phys. 18, 158 (2025)_

## Round 2 · Referee Report · Anonymous (Referee 2) · 2024-12-20

Report

In this article, the author considers the statistics (mostly the average) of the Euler characteristics for solutions of $M$ random constraint satisfaction problems $V_k({\bf x})=V_0$ where the vector ${\bf x}\in \sqrt{N}{\cal S}_{N-1}$ is on the $N$-sphere. This quantity provides interesting information on the topology of the solution space. The author considers specifically the limit $N,M\to \infty$ with fixed ratio $\alpha=M/N$. From the expression of the average Euler characteristics, the author obtains a phase diagram for the model, separating various topologies for the space of solutions. The results on the average are further confirmed and extended by computations of the second moment as well as the average logarithm of the Euler characteristics.

As a byproduct of the computations, the author derives interesting conjectures on spherical spin-glass model by taking the limit $\alpha,V_0\to 0$, with $E=\alpha^{-1/2}V_0$ fixed. For $M=1$, the solutions of the constraint satisfaction problem indeed coincide with level-sets of the spherical spin-glass.

The article is very well written and reports new and interesting results. I therefore recommend it for publication in SciPost Physics.

Please find below a short list of comments/ typos:

-p 10: "This fact mirrors another another that was made clear recently"

-Some of the results derived for the spherical spin-glass stem from the limit mentioned above. It is clear that this limit may be relevant as some important energy scales are recovered that way. Note however that starting from a regime $\alpha=M/N=O(1)$ and taking the limit $\alpha\to 0$ should in fact match with the limit $M\to \infty$ of the regime $M=O(1)$. Thus there is no guarantee that it should provide relevant information on the case $M=1$. A direct study of the level set of the spherical spin-glass model as done in Appendix D should allow to confirm this prediction. It was not entirely clear from the manuscript whether this analysis indeed does confirm the relevance of $E_{\rm sh}$.

-As mentioned by the author, the numerical data does not allow to judge the relevance of the energy scale $E_{\rm sh}$ and this direction deserves further investigation

Recommendation

Publish (meets expectations and criteria for this Journal)

  • validity: high
  • significance: high
  • originality: high
  • clarity: high
  • formatting: excellent
  • grammar: perfect

Author:  Jaron Kent-Dobias  on 2025-03-11  [id 5278]

(in reply to Report 1 on 2024-12-20)

Response to referee comments

  • We fixed this typo.
  • The question of limits is a shrewd one, but ultimately the result is the same no matter how the calculation is done. Working directly at M = 1, the steps in the appendices are followed up to equation (28). With M = 1 and V₀² = NE, the second term in the exponential remains of order N but the second is of order 1 and becomes another contribution to the prefactor. Comparing the resulting expression with (41) in the limit of α to zero with V₀² = E²/α, the two approaches result in the same effective action. In fact, an earlier version of this manuscript included two derivations, but the one for M of order 1 was deemed redundant in light of this. A note about this point has been added to the amended manuscript.
  • We agree, and further emphasized this by adding a sentence to the abstract of the amended manuscript.

---

## Round 2 · Referee Report · Anonymous (Referee 1) · 2024-12-20

Strengths

1- The work deals with a constraint satisfaction problem that was introduced recently, which serves as a toy model for both confluent tissues and non-linear regression in high dimension; it addresses a problem (the characterization of the topology of the solution space) that is of interest for this model, but also beyond.

2 - The author points out a limit in which the constraint satisfaction problem maps into well known mean-field models of glasses, allowing them to formulate a conjecture on the dynamical meaning of the shattering transition in this limit.

3 - The appendices contain some careful analysis (e.g. of lower order contributions which contribute to the refactor of the average Euler characteristics). The main text remains very readable, the content is well distributed between main and appendices.

Weaknesses

1 - The comparison with previous results in the literature and the discussion on the instability towards RSB phases could be clarified further (see comments below).

Report

The work characterizes the topology of the manifold of solutions of a high-dimensional, random constant satisfaction problem, which was introduced recently in the literature in connection to problems of confluent tissues and of non-linear regression in high dimension. The topology is characterized by computing the average Euler characteristics of the solution manifold, in the limit of large dimensionality N of the configuration space. The analysis allows to identify five distinct regimes as a function of the model's control parameter V0. These regimes differ by the magnitude of the average Euler characteristics (exponential in N vs order one in N), by its sign and by the regions of configuration space that contribute dominantly to it. A scaling limit is also discussed, in which this analysis corresponds to the characterization of the topology of level sets of the energy landscape of spherical p-spin models. Within this context, the author proposes a dynamical interpretation of the "shattering energy”, which corresponds to the energy level where a transition occurs between different topological phases, as being predictive of the dynamic threshold.

The manuscript is clear and self-contained. Below are some questions and requests for clarifications:

(i) It is argue in the paper that in the regime V<V_sh, where the action at m=0 is complex, that in the regime where the action at, the solution should not be discarded as it indicates a negative average Euler characteristic. This argument is supported by the calculation of the second moment and the equality (18). Regarding the m* solutions, the action at m* becomes complex above V_on: is there an interpretation for these solutions in the regime V> V_on?

(ii) It is mentioned that the naive satisfiability threshold predicted from the vanishing of (12) coincides with the threshold obtained within the replica symmetric analysis of the cost function (3). By reading the manuscript, I have missed if/how the satisfiability threshold arising from the analysis of the average Euler characteristics compares with the threshold obtained from the zero-temperature analysis of the equilibrium problem with energy (3): could the Author comment on this?

(iii) Related to (ii): a general discussion on the instability of the average Euler characteristics to RSB is presented in Appendix C2, leading to the prediction (79). In the case of the spherical models illustrated in Figures 3 and 4, this instability seems not to be relevant in the regions where the SAT-UNSAT transition occurs. It is not clear to me whether all the cases illustrated in the plot are such that the correct ground state is found within a simple RS formalism from the T=0 equilibrium calculations, or whether there is no relation between the RSB instabilities occurring within the two calculations. A comment on this could be added in the text.

(iv) If understand correctly, the vector x0 is arbitrary and it is introduced with the purpose of decomposing the contributions to the Euler characteristics in terms of m. Given the arbitrarily of x0, one would naively expect that the “observable” part of the solution space corresponds to m=0, and that any analysis of the constraint satisfaction problem that is x0-independent should be unable to pick up the transition between Regime II and Regime III: is this the case?

Requested changes

1- I would clarify the connection to previous results, if available, particularly in relation to points (ii) and (iii) mentioned above.

2- Possibly add some comments on the other points in the report.

3- Consider adding more information to Fig. 2, such as the notation for the values of the order parameters where transitions occur (V_on, V_sh, V_SAT), and the key features of the average Euler characteristics in each regime.

4- Include a comment stating that the functions in equation (7) are Morse, which justifies the use of equation (6). The Smale condition is mentioned, but not explained.

Recommendation

Publish (meets expectations and criteria for this Journal)

  • validity: -
  • significance: -
  • originality: -
  • clarity: -
  • formatting: -
  • grammar: -

Author:  Jaron Kent-Dobias  on 2025-03-11  [id 5279]

(in reply to Report 2 on 2024-12-20)

Questions and requests for clarifications

  1. In fact, the action is not complex when evaluated at m_* for V₀² > V_on² even though m_* itself becomes complex: the action remains real but becomes negative in this regime. This means that the contribution of these complex-m_* solutions in this regime shrinks with increasing N, and rather than representing a subleading but exponentially large (or even order 1) contribution to the Euler characteristic, their contribution is negligible.
  2. The reference "A continuous constraint satisfaction problem for the rigidity transition in confluent tissues", which performs the FRSB treatment of the zero-temperature equilibrium problem for the case where f(q) = ½ q² and α = ¼, estimates V_SAT ≃ 1.871. Our calculation instead predicts V_SAT = 1.867229…. In private correspondence with the author of the quoted reference, they indicated that such a discrepancy could easily be due to inaccuracy in the numeric PDE treatment of the FRSB equilibrium problem and that they were not concerned by the seeming inconsistency. So, for the moment the two treatments are consistent but the agreement is not precise. A small discussion of this has been added in a footnote to the manuscript.
  3. The irrelevance of RSB to the spherical spin glasses represented in the α → 0 limit of the included phase diagrams is expected. In both the pure spherical models (Fig. 3) and the mixed 1+2 models (Fig. 4) the equilibrium measure is always either replica symmetric or 1RSB, and the distribution of stationary points in both is always replica symmetric. However, the paper does include a discussion of the consistency between the RSB instability predicted by our second moment calculation and the appearance of RSB in the complexity of the spherical spin glasses, at the end of Appendix D (they are consistent). Not said in the manuscript is that this agreement also exists with the instability in the zero-temperature equilibrium measure and the satisfiability threshold, whose calculation is an intermediate step in finding the quenched shattering energy. If the referee is also curious about the agreement between RSB instabilities in the zero-temperature equilibrium treatment of the cost function when α > 0, we addressed this briefly in the final paragraph of Appendix C. There are regions of the SAT–UNSAT transition for the case f(q) = ½ q² where the equilibrium cost function is FRSB, where this calculation does not have an instability. As noted in that paragraph, there are reasons to believe that this is a trait of the cost function itself, since the cost function is predicted to have such an instability for a mundane energy level set of the pure 2-spin spherical spin glass where no RSB occurs (recall that for spherical spin glasses, the cost function is the square of the usual Hamiltonian).
  4. The picture described by the referee is partially true. The value of the Euler characteristic is independent of how x₀ is drawn, but this does not mean that elements of the calculation depending on m, the overlap with x₀, are unobservable. The simplest example is for the linear f(q) = q case, where for V₀² < V_SAT² the entire contribution to the Euler characteristic is made at m² > 0. The aspect that is malleable is at what value m_* the contribution is made. Since we draw x₀ uniformly on the sphere, m_* can be interpreted as the expected value of the overlap between a uniformly random point in configuration space and the nearest piece of the solution manifold. If x₀ were drawn in a different way, e.g., from a Boltzmann distribution on the cost function at finite temperature, then the value of the Euler characteristic computed would not change but the value of m_* would, and also our interpretation of its value. Whether such a change would modify the location of the onset transition V_on isn't known. A discussion of this issue has been added in a footnote to the amended manuscript.

Requested changes

  1. We have done this with respect to point (ii), but not (iii) where discussion already existed in the manuscript.
  2. Some comments have been made.
  3. At the moment when the manuscript is typeset Fig. 2 is on the same page as the description of the topological phases containing the requested information. Therefore, adding them to the caption feels redundant. However, if the referee feels strongly that the information should appear in both places the modification can be made.
  4. Such a comment has been added.

---

## Round 2 · Referee Report · Stefano Sarao Mannelli (Referee 3) · 2025-1-31

Strengths

1- The study addresses a gap in understanding the loss landscape of CS problems, a field that has been challenging due to technical complexity. 2- The application of the Kac-Rice formula to study Euler characteristics in high-dimensional settings is, to my knowledge, novel and simplifies the calculations considerably. This approach could potentially be extended to other problems where progress has been hindered by the sign of the determinant. 3- Technically, this remains a difficult problem, highlighting the significance of the result. 4- The results provide valuable insights, identifying a complex picture of the landscape and several regimes that may help elucidate the dynamics. 5- The final question addressed in the paper is significant, as it has remained elusive in previous studies. While I am not fully convinced by the author's argument (detailed later), it introduces valuable new elements to the discussion.

Weaknesses

1- The paper provides insufficient discussion of previous work. The author condenses key background and literature into two brief sentences (at the start of the second paragraph of Section 1 and at the end of page 2). Even for readers familiar with these references, this is hard to parse. I had to go into the bibliography and see which paper the author was referring to in order to follow. I strongly recommend expanding the introduction and discussing prior work in greater detail to provide adequate context for the reader. 2- The interpretation of magnetization m is unclear. While briefly mentioned at the beginning of Section 2.2, the explanation is insufficient. Since there is no planting in this problem, the physical meaning of an arbitrary random direction is still unclear to me. 3- Although the introduction to Euler characteristics in Section 2.1 is generally well-presented, I am still uncertain about some aspects: 3.1- At the beginning of Section 2.2, the author notes compatibility with an N−M−1 sphere, yet the Euler characteristic should be 2 for any hypersphere regardless of dimension. Can something be concluded about dimensionality here? 3.2- The large Euler characteristic could result from either many disconnected components or the manifold being a product of many manifolds, but the analysis does not distinguish between these cases. How might these scenarios lead to different landscapes? Can we say something about the possible implications for the dynamics? 4- The connection with the dynamics is not fully convincing. In particular, the theory provided in the paper does not explain the relationship between dynamics and the temperature dependence observed in references [26, 27]. These references identify different behaviours based on the initial temperature in a mixed p-spin model, yet this aspect does not seem to emerge. 4.1- Additionally, I wonder if the authors have considered how planting would affect the landscape. In the mixed p-spin case, planting simplifies the picture compared to what was observed in [26, 27]. 5- The derivation lacks sufficient detail in some sections. The author uses properties of the superdeterminant without providing references, making it difficult to follow. For example, the steps leading to equations (37-39) are unclear. 5.1- In equation (47), a superdeterminant with a suffix is introduced without a definition, which makes it challenging to interpret.

Report

This paper addresses a technically challenging and important problem, making a significant contribution to the study of loss landscapes in constraint satisfaction problems. The innovative application of the Kac-Rice formula and the identification of new regimes add meaningful insights to the field. Although there are some areas requiring clarification and expansion, especially regarding background context, the interpretation of the order parameter, and the connection to dynamics, these revisions mainly pertain to the clarity and depth of exposition rather than fundamental issues. Overall, I recommend acceptance, contingent on addressing the previously mentioned issues in a revised version.

Requested changes

1- Expand the introduction, providing more context for the problem and discussing relevant previous contributions. 2- Clarify the physical interpretation of the order parameter m. 3- Discuss the different scenarios that could result from a large Euler characteristic and their implications for the landscape and dynamics. 4- Provide further clarity on the connection with dynamics, specifically addressing the temperature dependence observed in prior work and the potential effect of planting. 5- Provide additional detail in the derivation, particularly in Sections A and B.1.

Recommendation

Ask for minor revision

  • validity: high
  • significance: good
  • originality: high
  • clarity: low
  • formatting: perfect
  • grammar: perfect

Author:  Jaron Kent-Dobias  on 2025-03-11  [id 5280]

(in reply to Report 3 by Stefano Sarao Mannelli on 2025-01-31)

Since the weaknesses and the requested changes coincide, we address both simultaneously.

  1. The discussion of the previous literature on this model has been expanded in the introduction, as well as some small contextualization as to the content of our evidence of generic interest in the topic on the first page.
  2. A discussion of how to interpret the order parameter m has been added to a footnote in section 2.1.
  3. See the comments below.
    • The referee is wrong to say that the Euler characteristic of a hypersphere is 2 independent of dimension. The Euler characteristic of all odd-dimensional manifolds is zero. Consider the cell complex on Spictured here. The Euler characteristic calculated using the alternating sum over the number of cells of increasing dimension is χ(S₁) = 1 – 1 = 0.
    • In this manuscript we present what we consider to be the simplest interpretation of the calculation, but the referee is correct to point out that a large Euler characteristic could indicate a complicated product manifold as well as one with many connected components, or other exotic manifolds besides. Our intuition for favoring an interpretation with many connected components is that applying one constraint amounts to taking a smooth, non-self-intersecting slice of a sphere, and repeating this many times feels likely to lead to unions of mostly spheres. This schematic argument has been added to the manuscript as a footnote in section 2.3. As to what dynamics might look like in a problem where the manifold of solutions were actually a nontrivial product manifold or something more exotic, we have no idea.
  4. The referee points out that previous work on gradient descent in the spherical spin glasses studied gradient descent from both uniformly random initial conditions ("infinite" temperature) and initial conditions drawn from a Boltzmann distribution at some finite temperature, and found that the final state of the dynamics reached minima in a range of energies depending on the initial condition. The conjecture in this manuscript seeks only to explain the upper energy of this range, that associated with gradient descent from a uniformly random initial condition. Presumably there are a variety of behaviors observable by choosing initial conditions using a variety of initial distributions, Boltzmann or otherwise, and one day we may hope to address such questions using similar approaches to this paper. However, this is not addressed here. A small discussion of this point has been added to the manuscript.
    • A paragraph addressing what might occur in planted models has been added to the manuscript.
  5. The existing citations to references regarding the use of superspace coordinates and operators have been clarified in the new manuscript, including an explicit reference to an explanatory appendix on the method written by the author. Repeating the same content here seems unnecessary. The relationship between the right and left-hand sides of (37–39) were made using the elementary rules outlined in the aforementioned appendix and symbolic algebra software, with no other intermediate steps to share.
    • The subscript notation associated with the determinant has been explained in a footnote.

---

## Round 3 · Referee Report · Anonymous (Referee 1) · 2025-3-30

Report

The Author has address the comments of my previous report, and made appropriate changes in the revised version of the manuscript.

Recommendation

Publish (meets expectations and criteria for this Journal)

---

## Round 3 · Referee Report · Stefano Sarao Mannelli (Referee 3) · 2025-4-8

Report

The author addressed the concerns of my previous report and updated the manuscript accordingly.

Recommendation

Publish (meets expectations and criteria for this Journal)

---

## Round 3 · Referee Report · Anonymous (Referee 2) · 2025-4-16

Report

The author has addressed all of the referees' comments in a satisfactory manner.

Recommendation

Publish (meets expectations and criteria for this Journal)

---

## Round 3 · Author Response

We thank the referees for their careful reading of the initial manuscript and for their helpful comments. The revised manuscript has benefited from changes attempting to implement their advice and address their concerns. Please see the responses to the individual reports for more context regarding the changes.

---

## Round 3 · List of Changes

A latexdiff of the changes since the first submission is available here: https://kent-dobias.com/files/2409.12781v2-v3_diff.pdf

An itemized list of changes follows.

  • Affiliations, contact email, and funding information were updated.
  • A sentence was appended to the abstract emphasizing the imprecision of the evidence supporting our conjecture.
  • More context was added to references in the introduction, and some references were added.
  • A comment was added referencing that H must be a Morse function.
  • A footnote was added explaining the interpretation of the order parameter m.
  • A footnote was added explaining the agreement of the satisfiability threshold computed here and that computed using the zero-temperature equilibrium calculation.
  • A footnote was added explaining our choice to favor a specific interpretation of a large Euler characteristic.
  • Some minor mistakes were fixed in the beginning of section 3.
  • A footnote was added explaining the commutation of limits N → ∞ and α → 0.
  • Paragraphs were added discussing the relationship of our dynamic conjecture with other algorithms and with problems with a deterministic piece.
  • The references regarding the use of superspace coordinates were given more introduction.
  • A footnote was added to explain subscript notation of the determinant.

---

## Editorial Decision

published